# Prefrontal modulation of striatal engagement: a hierarchical framework for goal-directed behavior
Aron Koszeghy ✉ & Johannes Passecker ✉

We review Prefrontal-striatum interplay in goal-directed behavior, focusing on prefrontal (PFC) and striatal macro- and micro-domain interactions in hierarchical engagement with goals, strategies and actions. This dual-control framework posits a role for the striatal matrix in engaging with behavioral options- for automatization, and prefrontal modulation of striatal engagement for flexibility. Hierarchical prefronto-basal-ganglia loops incorporating ventral-, dorsomedial-, and dorsolateral striatum represent goals, strategies and actions respectively. The striatal "switchboard" enables engagement, a unified function across macro-domains, while its modulation by PFC operates through dual mechanisms: fast switching via direct matrix inputs, and slower, learning-based adaptation using a striosome-dopamine pathway.

**The prefrontal cortex (PFC) and the striatum, the main input nucleus of the basal ganglia (BG), are crucial for various aspects of goal-directed behavior[1–3]. While these regions have been extensively studied in isolation, understanding how they collaborate to drive flexible decision-making remains a challenge. Historically, this gap was due to methodological limitations, but recent advancements such as multi silicon probe recordings, cell type-specific expression methods and optogenetics have begun to illuminate these complex interactions[1,4–6]. A comprehensive understanding requires considering the prefrontal and striatal circuits not as isolated units, but as parts of an integrated system that considers most, if not all, of the prefrontal and striatal subdomains at the same time.**

This review focuses on prefrontal and striatal areas with shared homologs in rodents and primates (see Table 1 for a comprehensive list). In the rodent, the PFC includes the prelimbic (PrL, A32), anterior cingulate (ACC, A24), medial- ventral- and lateral- orbitofrontal (m/v/lOFC, MO A14, VO A13a, LO A13l/m), and infralimbic (IL, A25) cortices[7–9]. The striatum is functionally divided into three macrodomains: the dorsomedial (DMS), ventral (VS), and dorsolateral (DLS) striatum. Internally, these macrodomains are further compartmentalized into striosomes (patches) and the matrix. Another layer of striatal organization involves the distinction between direct pathway (D1R-MSN) and indirect pathway (D2R-MSN) medium spiny neurons (MSN). In coordinated opposition, the direct pathway facilitates the selected, fitting, action (behavioral option), while the indirect pathway suppresses competing alternatives; with their relative balance determining together the engaged behavioral option[10–12]. Both cell types are found within striosome and matrix compartments[13,14], in differing proportions and reviewed elsewhere[15–21]. While we focus on the striatum, the largest input nucleus of BG, we briefly need to acknowledge the importance of other nuclei of BG. The Subthalamic Nucleus (STN), as the stand-alone input nucleus of the hyper direct pathway, has multiple suggested functions (which can supplement behavioral option engagement); including fast global inhibition of engaged behavioral options allowing post-hoc re-engagement with others[22,23]. The output nuclei (globus pallidus internus—GPi, Substantia Nigra pars reticulata—SNr) of BG, provide constant inhibition to their thalamic and brainstem targets, keeping the whole repertoire of behavioral options inactive (except the ones matching current situation). As cortical areas and the behavioral repertoire of different species change, the structure and relative importance of BG nuclei and sub-nuclei also change[24], including species-specific differences in the topographic organization of GPi and SNr projections to thalamic nuclei involved in PFC-BG loops[25,26].

Our analysis centers on goal-directed behavior (often used interchangeably with "model-based" behavior), which is characterized by its deliberate and informed nature. It is distinguished from habitual behaviors by key features, including sensitivity to current outcome value[27] and the capacity to generate novel action sequences to achieve a desired outcome[28]. Crucially, it relies on an "internal world model" representing the causal structure of the environment, allowing agents to predict consequences and guide their actions[3,29–31].

To examine the interplay between the PFC and striatum in supporting goal-directed behavior, we propose an integrative framework built on five core concepts: i) Hierarchical Selection: Goals,

Institute of Systems Neuroscience, Medical University of Innsbruck, Innsbruck, Austria. ✉e-mail: aron.koszeghy@i-med.ac.at; johannes.passecker@i-med.ac.at

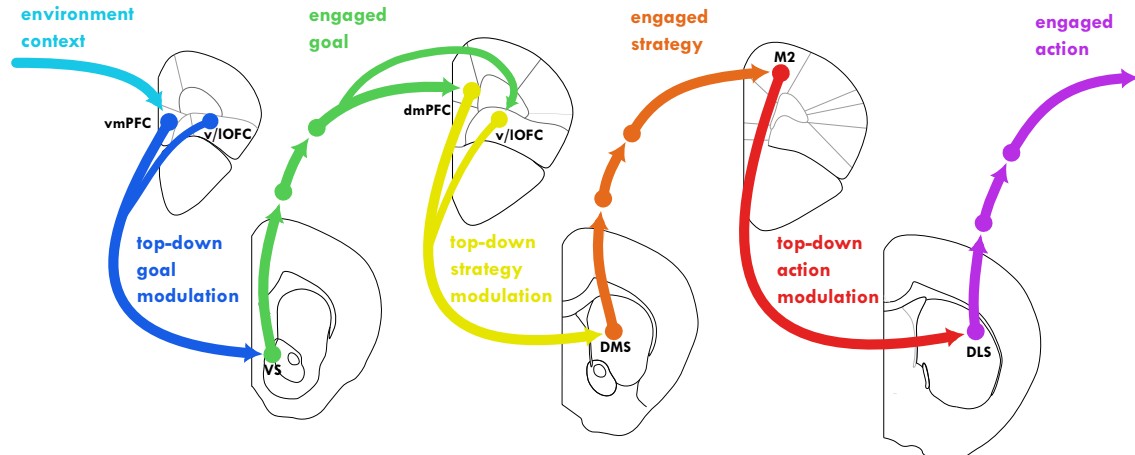

**Fig. 1 | Hierarchical Selection and Engagement with Goals, Strategies, and Actions via Prefrontal cortex Basal Ganglia (CBG) Loops.** This figure illustrates (from left to right) how hierarchical engagement with goals, strategies, and actions is achieved through serially connected CBG loops. Each striatal macrodomain region receives topographically organized input from specific cortical areas. Through the output nuclei of the BG and thalamus, these striatal regions project non-exclusively back to their respective cortical input areas forming CBG loops. The sequential flow of information through these interconnected loops supports hierarchical striatal engagement with multiple behavioral options. Note that the two rows represent the prefronto-striatal dual-control over behavior, upward arrows pointing away from the striatum represent behavioral-engagement, downward arrows pointing away from PFC represent top-down modulation of engagement. To emphasize the information flow between loops, the recurrent connections returning to the originating cortical regions are not shown. Brain illustrations are based on The Allen Mouse Brain CCF[271,272,273]. vmPFC stands for ventro-medial prefrontal cortex, v/lOFC stands for ventral and lateral orbitofrontal cortex (see also Table 1 and main text).

strategies, and actions are selected and engaged concurrently via serially connected PFC-BG loops. ii) Unified Matrix Function: Across all macrodomains, striatal matrix ensembles function to maintain engagement (temporal commitment) with a selected behavioral option. This mechanism allows for the automatization of specific behaviors, whether goals, strategies, or actions, without requiring continuous attention during stable periods. iii) Associative Learning: Learning is enabled across all striatal macrodomains by two key features: the high convergence of synaptic inputs from various cortical and subcortical (including thalamic) areas onto matrix MSNs, and the capacity for dopamine-dependent plasticity at these synapses. iv) Evaluative Learning Loop: The PFC influences learning by providing evaluative input to striosomes. In turn, striosomes regulate midbrain dopamine neurons, which modulate plasticity within the matrix. v) Real-time Switching: Direct PFC synaptic inputs to the matrix exert real-time control over behavior. These inputs can either initiate engagement by activating a specific learned association (an MSN ensemble) or rapidly switch activity to a different ensemble, thereby determining the currently selected and engaged behavioral option.

By integrating these concepts, we outline a theoretical model (Fig. 1) in which the PFC and striatum cooperate continuously to balance stable engagement with flexible adaptation. We critically evaluate the evidence supporting this integrated model and highlight key areas for future research.

## A dual-control architecture for automatization and flexibility

What is the role of prefrontal-striatal communication in goal-directed behavior? We propose that it facilitates optimized engagement with behavioral options. We derive this terminology from Hierarchical Reinforcement Learning[32] as a closed-loop control policy that extends over time. In this framework, selecting a behavioral option implies engaging with it, maintaining its execution until a specific termination condition is met. This applies hierarchically to actions, strategies, and goals. This is in line with, and extending beyond the motoric domain, of the basal ganglia (BG), and within that striatum (its primary input nucleus), as a programmable 'switchboard' that represents the entire action space[33–35]. The

striatum's primary role is not just to initiate actions, but to select and maintain engagement with specific behavioral options, thereby regulating their onset, duration and offset[34,36–39]. This selection occurs predominantly via cortico-striatal and thalamo-striatal synaptic input patterns activating striatal MSN ensembles, which in turn represent engagement through disinhibiting a chosen behavioral option from a pool of competing, inhibited alternatives[40–45].

The PFC modulates this striatal 'switchboard' to initiate behavioral adaptation when environmental state changes necessitate it. This is achieved through two primary, dissociable pathways[1,5,30,46,47].

1) Slower Learning Pathway: PFC projections to striosomes provide evaluative signals[46]. These striosomal projections, via their connections with midbrain dopaminergic neurons[48], can indirectly modulate cortico-striatal and thalamo-striatal synaptic plasticity in the matrix[49–51] (see also later sections), enabling striatum to learn, re-learn and adapt via re-routing associations between stimuli, actions, and outcomes[52–54]. We define this process as slow because it relies on dopaminergic synaptic plasticity (reinforcement learning) that is induced and expressed over minutes to hours, necessary for consolidating new associations.

2) Faster Switching Pathway: Direct PFC projections to the striatal matrix MSNs[14] or interneurons[55], convey information about the current state of the internal world model, enabling rapid, model-based switching between engaged behavioral options[3,30,31,56,57]. This is a faster process as it operates on the timescale of synaptic transmission and population dynamics (milliseconds to seconds). This allows for real-time flexibility, such as switching between pre-established strategies on a trial-by-trial basis, without the immediate need for synaptic plasticity. A potential supplementary mechanism for rapid behavioral inhibition is the cortico-subthalamic hyperdirect pathway, which can operate in parallel to these striatal circuits[58,59].

This dual prefronto-striatal behavioral control architecture allows for both computationally efficient, automatized behavior when adjustments are not required, and enhanced behavioral flexibility when they are.

This automatization is expected to not be limited to the well established stimulus-response (S-R) habits of the DLS[60–65]; wherein, once associations are learnt, specific patterns of cortico-striatal and thalamo-striatal inputs can elicit motor actions by activating corresponding DLS MSN ensembles[66–72], bypassing the need for conscious deliberation. Rather, we

predict that the underlying neurobiological mechanism, associating specific input patterns with specific MSN ensemble activity, extends across all striatal macrodomains and thus likely to other domains of the extended behavioral-option space[73]. Consequently, engagement is a unified function that applies hierarchically to goals, strategies, and actions (Fig. 1), which are implemented via interconnected CBG loops[74–80]. Anatomical evidence shows these loops are not only recurrent via the thalamus but also connect to other cortical areas[78,81–83], creating the substrate for hierarchical sequential selection of multiple behavioral options, and maintaining concurrent engagement with them. Such hierarchy consists of three layers: i) Goal Engagement: A ventromedial PFC (vmPFC)-VS loop selects and maintains engagement with a goal based on the current context[84]. Here, the term "goal" refers to a desired future state that an organism is actively pursuing, dependent on an anticipated outcome state itself (not for example a distinct spatial location delivering the outcome). ii) Strategy Engagement: A dorsomedial PFC (dmPFC)/lateral OFC-DMS loop then selects and engages with the optimal strategy to achieve that goal[1,4]. Strategy is defined here as an association between a desired outcome and outcome-related context with either an outcome-predictive stimulus, or an outcome-predictive response-type. These associations guide behavior by specifying which response-type or sensory stimulus to select given an engaged outcome, and specific environmental/task contexts, to achieve the desired outcome. iii) Action Engagement: Finally, a motor cortices/dorsal ACC-DLS loop selects and engages with the action that implements the chosen strategy[85]. Actions are simple motoric movements or responses that involve muscle contraction and can be composed of smaller action syllables.

This anatomically serial selection and concurrent engagement is crucial because goals, strategies, and actions require different timescales for planning and execution. While predictive models of varying complexity are necessary for representing goals, strategies, and action sequences[86–89], we would argue that these loops support both the selection and sustained engagement with these behavioral options. With modular expansion it allows the complexity of these CBG loops to vary between species based on their environmental demands and behavioral repertoire needs[90]. The advantages of nested, hierarchical behavioral automatization becomes particularly evident when pursuing complex goals that require multiple subgoals and actions[73]. This provides a crucial layer of cognitive offloading between high-level planning and action execution. Consciously attending to all of these elements over extended periods would demand significant, maybe at times unattainable cognitive effort. For instance, an experienced driver navigating a familiar route requires less cognitive effort than a novice driving the same route, illustrating the distinction between automatic and conscious control, even when the overall goal remains consciously chosen. Other forms of behavioral automatization, distinct from simple S-R habits, which are not an integral part of the engagement framework suggested here but can be viewed as complementary, have been reported earlier: for instance, "automation of willing", where frequently pursuing a goal in specific contexts leads to automatic goal activation upon encountering those contexts[91]. Another example is, "goal-dependent habits", where habitual actions are mentally represented as goal-action links that are automatically triggered only when the associated goal is engaged[92,93]. Both forms of automatization maintain sensitivity to outcome devaluation, reinforcing the principle that automaticity need not eliminate goal-dependence or functional adaptability; and both mechanisms can be combined in a chain with elements of the suggested three-loop engagement framework.

## Separate striatal macro domains support engagement with goals, strategies, and actions

*Which striatal macrodomains support engagement with goals, strategies, and actions? The VS, DMS, DLS* are hypothesized to be the anatomical substrates within the striatum responsible for engagement with *goals, strategies and individual actions* respectively[84,94,95]. The VS may represent goals, the DMS

the strategies required to achieve those goals, and the DLS the specific action sequences that comprise a behavioral response (dictated by the currently engaged strategy). Of note, we refer to the rodent macrodomains as the DMS is often considered analogous to the primate caudate nucleus, the rodent DLS to the primate posterior putamen, and the rodent VS to the primate nucleus accumbens[96]. The division of the striatum into three macro-domains is a simplification, other naming conventions based on cortical inputs offer higher anatomical resolution[97,98]. The striatum is characterized by a gradient of cortical and thalamic synaptic inputs shifting from ventromedial to dorsolateral regions[99]. This gradient likely supports engagement with behavioral options across shrinking timescales; from long-term goals in ventro-medial regions to strategies and finally millisecond-scale action syllables in dorso-lateral regions[100]. Despite this, we adhere to the standard three-domain nomenclature in this review for two reasons. First it aligns well with the conceptual distinction between goals, strategies, and actions, and second because the bulk of the existing literature uses this terminology, limiting our ability to map functions to more precise sub-regions.

### Ventral striatum

The proposed role of the VS and the most ventro-medial part of DMS (vmDMS the regions which are strongly innervated by vmPFC projections[101,102], see also Table-1, projection strength is defined in these anterograde tracing studies based on qualitative metrics of presynaptic fiber and terminal density) in selecting and maintaining engagement with goals provides a unifying framework for many of its classically associated functions. For example, electrical, pharmacological or optogenetic stimulation of the VS in humans and animals enhances incentive salience[103–105], the motivational wanting attributed to goals and their associated cues, and can induce self-stimulation behavior[106–112]. This increased motivational salience is consistent with sustained goal engagement, as stronger engagement with a specific goal would naturally lead to increased motivation to approach or achieve it. The following reports from human VS DBS subjects illustrate how very specific and complex goals may be activated by stimulating this brain region[109]. "Almost immediately (60 s) after switching the stimulation on, one patient was unable to identify any changes, but spontaneously reported that he realized that he was in Cologne, that he never visited the famous Cologne Cathedral, and he planned on doing this in the immediate future, which he indeed did the day following the operation. A second patient's immediate (60 s) reaction to stimulation was quite similar; she did not report any acute changes in depressive symptomatology but spontaneously mentioned that she wished to take up bowling again (a favorite pastime activity of hers from 12 years before)". Unlike in human studies, goals in animal experiments must be inferred from choice behavior. Nevertheless, research has demonstrated that ventral striatum activity is necessary for initiating and re-initiating goal pursuit in rats[113,114]. Furthermore VS activity represents predicted outcomes in rats and primates[115–117], and predicts or precedes behavioral engagement with goals in rats[103,118,119].

Such reports further suggest that VS stimulation promotes engagement with specific, long-term goals, not just aspecific reward seeking. Similarly, the VS's role in addiction[120,121] can be viewed as a pathological narrowing and intensification of goal engagement by prioritizing a drug-induced outcome state at the expense of other goals. VS lesions impair learning driven by changes in reward value[3], potentially indicating its role in goal-engagement and in updating goal representations.

Finally, beyond connections to the midbrain dopaminergic system the VS's connections to the ventral pallidum, substantia nigra pars reticulata, and through them ultimately to multiple PFC regions[122–124], provide an anatomical substrate for communicating goal engagement to other brain areas, influencing downstream strategy and action selection.

### Dorsomedial striatum

The DMS is proposed to be involved in the selection of, and sustained engagement with, strategies to achieve selected goals. Inactivation of the

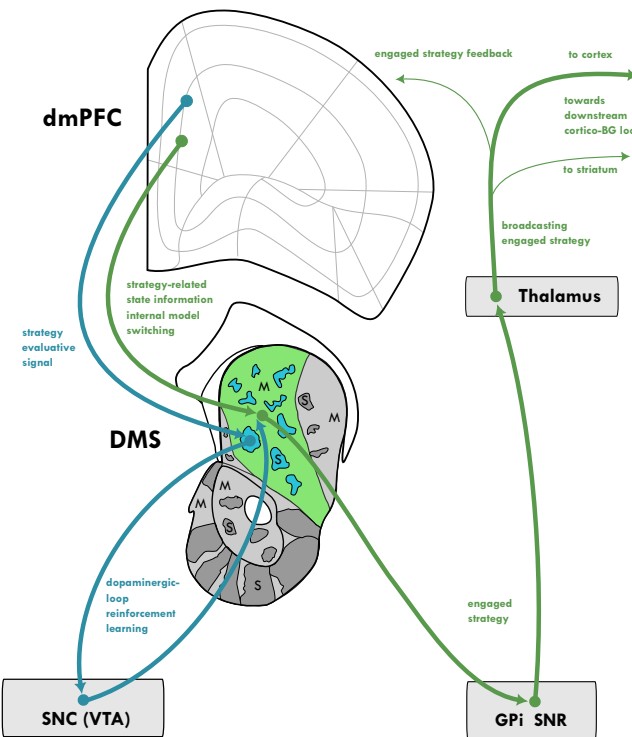

**Fig. 2 | Fast and Slow Prefrontal Modulation of Striatal Strategy Engagement via Matrix and Striosome Pathways in goal-directed behavior.** When environmental contingencies are stable, active matrix MSN populations represent the currently engaged strategy. The PFC modulates this engagement through two primary mechanisms: (1) Direct Matrix Modulation: The PFC sends information about the current internal model state directly to matrix MSNs. This input can rapidly alter the active MSN population, leading to a switch in the engaged strategy. (2) Striosome-Mediated Learning: The PFC provides model-based reward prediction information to striosomes. This input, relayed via the dopaminergic system, modulates synaptic plasticity in the matrix, facilitating long-term learning and adaptation of strategy selection. This figure represents the middle hierarchical loop from Fig. 1, strategy selection and engagement; here, only key brain regions involved in probabilistic decision-making are shown (in the absence of sensory cue instructions about strategy). While appearing as patches in 2D sections, striosomes should be conceptualized as a 3D network of pipelines.

DMS impairs both reversal learning and strategy switching[125], demonstrating its critical role in adapting strategies based on feedback. DMS neurons represent action-values[100] and advantageous strategies[6] in probabilistic decision-making tasks, suggesting that the DMS encodes information necessary for choosing and maintaining a particular strategic approach. Furthermore, dorsal striatal neurons, including those in the DMS, exhibit firing rate correlations with stimulus-value in classical and instrumental conditioning paradigms[126,127] indicating a role in learning and representing the value of different sensory cue based strategies. Recent evidence suggests a functional specialization within the DMS along the anterior-posterior axis, with the posterior DMS involved in learning and committing to strategies based on positive outcomes, and the anterior DMS involved in utilizing the lack of expected reward, potentially promoting switching to new strategies[2,128]. Further research is needed to determine whether this anterior-posterior specialization reflects a mechanism for regulating engagement with individual strategies or represents a fundamentally different role for the anterior DMS.

### Dorsolateral striatum

In goal-directed behavior, the DLS is proposed to be responsible for the engagement with actions in the correct order and duration, which can be dictated by a currently engaged rule or strategy[95]. DLS neurons systematically encode information about the identity, timing, duration and ordering of sub-second action syllables that form actions. Lesioning of the DLS prevents appropriate action-syllable-order in sequences[34]. DLS MSN ensembles encode action identity independently of movement speed[33], suggesting that the DLS represents the selected action itself, rather than how it is being executed. MSN ensemble activity represent actions, in a way that multiple populations of MSNs –including cells which change their activity throughout the duration of commitment[129–131] and also cells which increase their activity at the onset or offset of it[131,132] –together determine the time of action engagement. Thus, the DLS, as a whole can be viewed as a switch-board representing the entire space of available actions[33].

## Matrix and striosome micro-domains support different functions

What are the hypothesized functions of striatal microdomains in goal-directed behavior? Within each striatal macrodomain, microcompartments known as striosomes (or patches) and the surrounding matrix[20,133–135] exhibit distinct histochemical properties and connectivity, and thus are hypothesized to serve specialized functional roles (Fig. 2). Only MSNs from the

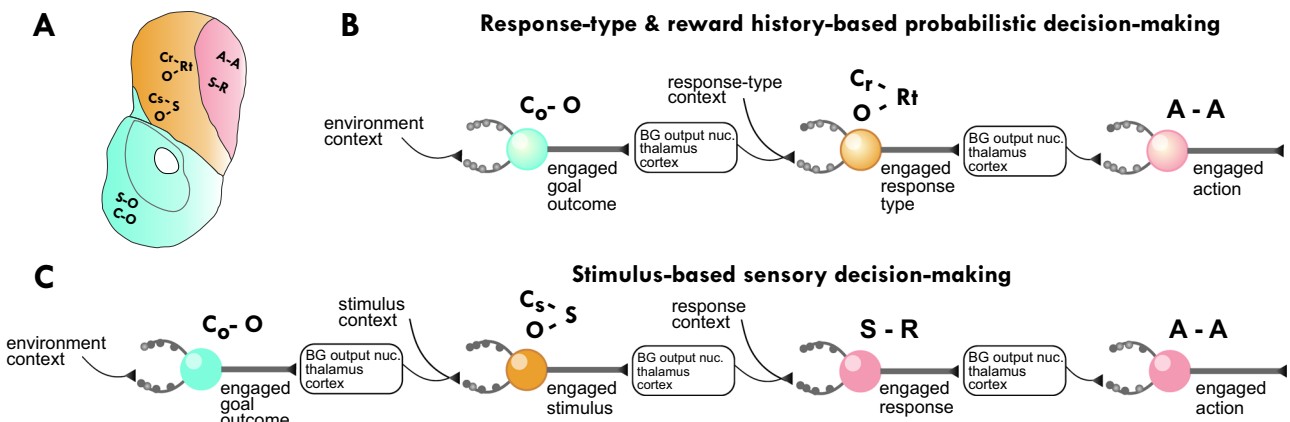

**Fig. 3 | Distinct Associations within Striatal Macrodomains Mediate Engagement with Specific behavioral options.** Learning, occurring at dendritic synapses and spines, modifies which inputs drive effectively which MSN ensemble. The activity of these MSN ensembles represents the currently engaged behavioral options, enabling automatization. From a comprehensive list of learned associations, within each striatal macrodomain (**A**); the sequential selection is depicted in (**B**) for response type and reward history based probabilistic decision-making, and in (**C**) for goal-directed decision-making driven by sensory cues.

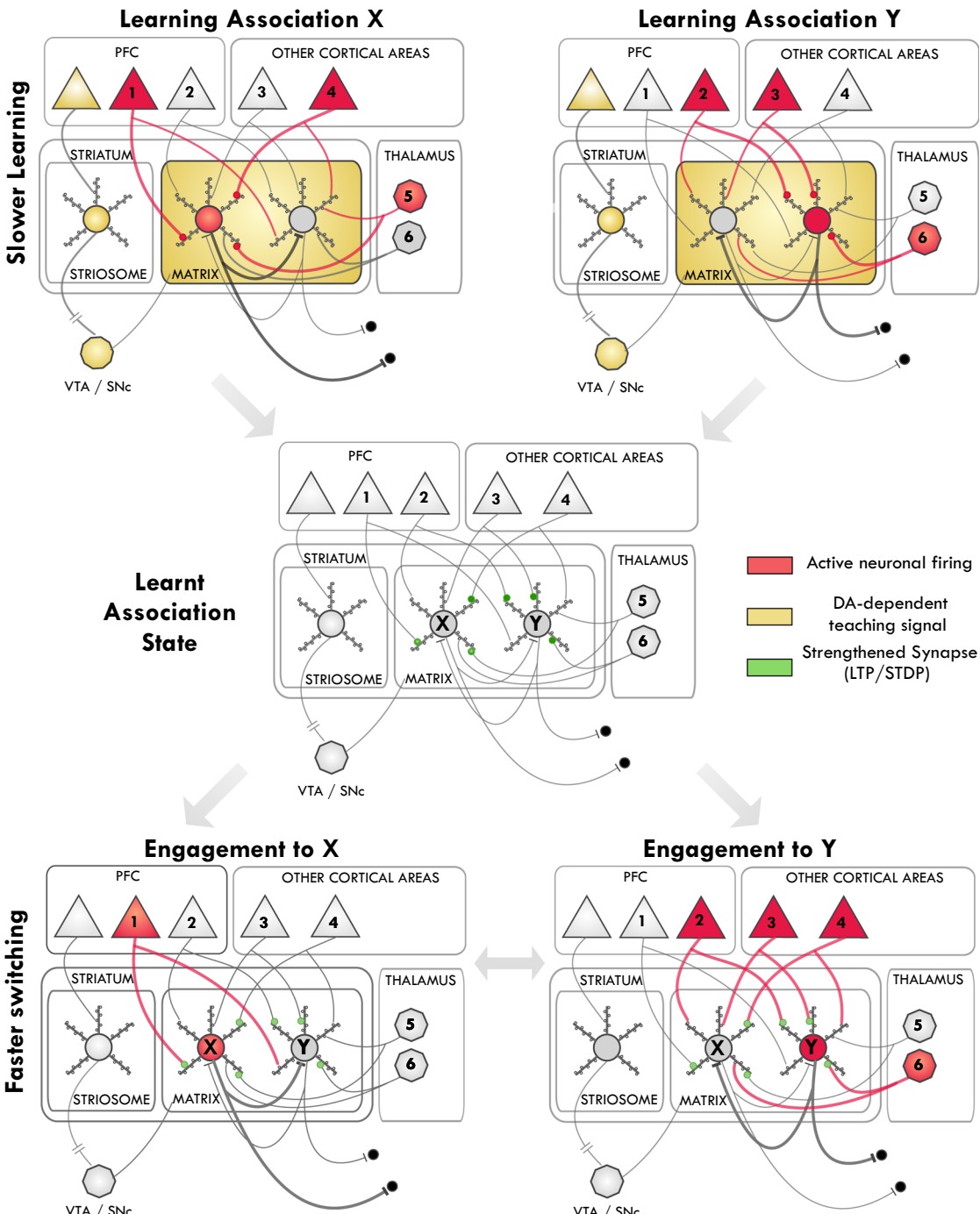

**Fig. 4 | Circuit Model for Prefrontal Modulation of Striatal Engagement through Learning and Switching.** In goal-directed behavior PFC modulates striatal function through two primary mechanisms: striosome-mediated slower learning (*Top and Middle rows*) and faster switching through direct matrix modulation (*Bottom row*) once relevant associations are represented. Striosome-Mediated Learning (*Top and Middle rows*): Learning is governed by a three-factor rule based on spike time dependent plasticity: synaptic connections onto matrix MSNs are strengthened when presynaptic input, postsynaptic MSN firing, and striosome-mediated phasic DA (yellow) release (representing a positive RPE) occur concurrently. (note: The mechanisms may differ between D1- and D2-receptor expressing MSNs). (*Top Panels*): The process of learning two distinct associations is shown. On the left, inputs from PFC/thalamic ensembles 1, 4, and 5 drive activity in matrix MSN ensemble X during a DA-dependent teaching signal, initiating plasticity. On the right, a different input pattern from ensembles 2, 3, and 6 drives another MSN ensemble Y, initiating learning of a separate association Y. (*Middle Panel*): The "Learnt Association State" depicts the result of the learning processes.

Enlarged green dendritic spines represent the strengthened synaptic connections (LTP/STDP), establishing a new input-output mapping, or "striatal routing." Direct Matrix Modulation and Behavioral Switching (*Bottom panel*)**:** Once associations are learned, the PFC can rapidly switch between them by altering direct input to the matrix. (*Bottom Left*): A specific behavioral option "element of the extended behavioral-space" is engaged as input from PFC ensemble 1 selectively activates the now-potentiated MSN ensemble X. (*Bottom Right*): To switch outputs, a different pattern of cortical/thalamic input (ensembles 2, 3, and 6) activates MSN ensemble Y. Although ensemble 4 provides competing input favoring ensemble X, the stronger convergent input to ensemble Y allows it to win the competition. This winner-take-all process, facilitated by lateral inhibition between MSN ensembles, ensures the selection of a single, dominant behavioral option. This illustrates how dynamic changes in cortical and thalamic input can rapidly shift engagement between established behavioral representations based on the learned synaptic strength profile.

## Table 1 | Function and Striatal Projections of Analogous Prefrontal Areas in Rodents and Primates

| Primate/Human | Rodent | Homology Evidence | Function | Main Striatal target |
|---|---|---|---|---|
| BA-10 (frontopolar cortex) | None - lateral FPI, is unique in humans | NA | Linked with meta cognition, social monitoring, managing multiple competing goals, multi tasking, cognitive branching, abstract analogical reasoning, a greater capacity to integrate temporally or semantically distinct information, a greater capacity for abstraction [274-278] | Cd - most medial part[279] |
| BA-11 - anterior OFC | None - limited functional analogy to rodent area14 mOFC | NA | Linked with outcome prediction, subjective value encoding, comparisons of outcome options, representing chosen value (potential functional links with BA-14, mOFC) [214,215,280,281] | Cd - most medial part[279] |
| BA-14 caudal; mOFC | MO; ORBm; medial orbitofrontal cortex | Conserved agranular/dysgranular cytoarchitecture. Both project heavily to VS and medial caudate, overlapping with IL/PL projections (IL/A25, PL/A32) [8,101,282] | Primate mOFC/vmPFC is critical for value comparison and goal selection. It encodes integrated value signals and is required for decision-making that is independent of irrelevant or distracting alternatives. Rodent data show MO involvement in goal-directed action, particularly when value must be inferred or recalled, although studies yield mixed support for MO vs. lOFC dominance in economic choice tasks [283-286] | VS & vmDMS[101,102] |
| BA-25 ventral; Subgenual; sACC | IL; ILA; A25; infralimbic cortex | Conserved agranular cortex region; overlapping projections to Nucleus Accumbens Shell as well as BLA, hypothalamus and periaqueductal gray [7,9,101,287,288] | Linked with prolonged motivational aspects related to task progression, encodes behavioral outcomes during goal-directed actions; likely controls extensively-trained behavior regardless of whether it is goal-directed or habitual [188,330,331] | VS & vmDMS[101,102] |
| BA-13a - Posterior Central OFC; BA-13 caudal medial part | VO; ORBvl; ventro-lateral orbitofrontal cortex; | Both rodent VLO and primate Area 13a are agranular and shared unique projection patterns[282] | In combination provides detailed Stimulus-Outcome (S-O) association information based on desirability and availability; Limited functional evidence on identified subregions [184,214,291-294] | BA-13a -> VS; BA-13l, BA-13m -> DMS [279,295] |
| BA-13l, BA-13m | LO; ORBl; lateral OFC; lateral orbitofrontal cortex | Rodent lOFC is agranular to dysgranular. Posterior Area 13 in primates is similarly dysgranular; strong reciprocal connections with BLA, and projections to the central striatum[8,279,282] | | |
| BA-32 - Pregenual ACC | PrL; PL; A32; prelimbic cortex | Conserved agranular cortex region; both rodent and primate area 32 projects to ventral striatum and ventro medial parts of the dorsal striatum [7,9,101,287,288] | vPrL is primarily linked with encoding of internal value representations and prolonged motivational factors; dPrL is indispensable for action outcome association based goal-directed behavior, promoting actions based on anticipated results. It is critical for Strategy Selection and Behavioral Flexibility in dynamic or ambiguous environments [56,224,286,289,290,297,298] | VS & vmDMS[101,102] |
| BA-9/46 - dlPFC | None - functional analogy to rodent area32 PrL; see above | NA | Working memory, cognitive control, executive function, abstract reasoning, rule representation [179,299-301] | Cd – dorsal part[302] Pu – medial part[302] |
| BA-47/12 - lateral OFC, vlPFC | None - (based on locational and functional similarities link to rodent LO | NA | Linked with stimulus outcome association, credit assignment, outcome availability representation, reversal learning [291,303-306] | Central striatum[279] (Cd & Pu) |
| BA-24 - ACC, ACAv | Cg2; 24a; A24a; (anterior cingulate cortex, ventral & anterior parts) | Support for homology with subregional spatial complexity [7,9,101,287,288] | Linked with Action-Outcome contingency degradation and with rapid adaptations to rule switches [307-310] | DMS & VS[101,102] |
| BA-44/45 - vlPFC, IFG | None | NA | Linked with gestural communication mirror neurons, and the generation of spoken or written language [311-313] | central striatum (Pu & lateral Cd)[314] |
| BA-24c, BA-a24c' - cMAr, MCC, ACC most dorsal & posterior parts | Cg1; 24b; A24b; ACAd; 24'; 24b';(cingulate motor areas, A most dorsal & most posterior parts) | Support for homology with subregional spatial complexity: there are multiple subdivisions within both rodent and primate ACC and MCC [7,9,101,287,288] | Linked with encoding the identity of the upcoming choice, implementation of trial-by-trial action selection, response selection for appropriate actions based on internal intentions and translating and updating rules into motor responses [188,296,315,316] | DLS & DMS[279,302] |
| BA-6 - multiple premotor areas including premotor cortex, SMA, pre-SMA | M2; MOs; secondary motor cortex | Support for homology with spatial complexity. rodent M2 contains multiplepremotor areas (ALM, aM2, pM2); that correspond to different primate BA6 subdivisions [317] | Critical for Action Selection and Motor Planning, acting as the interface that transforms spatial and rule-based information into executed movements [85,226-228,318] | DLS[98,319] |
| BA-4 - Primary Motor Cortex | M1; MOp; primary motor cortex | Conserved agranular cytoarchitecture, presence of giant Layer V pyramidal Betz cells, somatotopic organization, direct corticospinal projections [320] | M1 neurons strongly represent low-level motoric movement primitives, including both muscle activation and direction of movement in space and is involved in multiple stages of sensorimotor transformation including its timing [321-324] | DLS[98,319] |

While this review focuses on prefrontal and striatal regions conserved across species, this table also includes primate-specific PFC regions for context. It is important to note that while anatomical atlases traditionally often define sharp boundaries useful for simplified models, biological organization in both the PFC and striatum often follows functional and connectivity-based gradients that span these borders[99,188,190]. Here, Brodmann Areas are organized into three broad functional domains. Rodent terminology incorporates nomenclature from the Paxinos, Swanson, and Allen atlases. Color hues denote Goal(blue), Strategy (pink) and Action (gray) selection and engagement related larger anatomical units, which in rodents map on to vmPFC (& v/lOFC), dmPFC (& v/lOFC), and motor cortices, respectively. *VS* Ventral Striatum, *DMS* Dorsomedial Striatum, *DLS* Dorsolateral Striatum; *S-O* Stimulus-Outcome, *A-O* Action-Outcome, *OFC* Orbitofrontal Cortex, *ACC* Anterior Cingulate Cortex; *MCC* Midcingulate Cortex, *BLA* Basolateral Amygdala; *NAc* Nucleus Accumbens, *PAG* Periaqueductal Gray, *vmPFC* ventromedial Prefrontal Cortex, *dlPFC* dorsolateral Prefrontal Cortex, *vlPFC* ventrolateral Prefrontal Cortex, *IFG* Inferior Frontal Gyrus, *SMA* Supplementary Motor Area, *cMAr* rostral cingulate motor area, *vmDMS* Dorsomedial Striatum (most ventral and medial aspects), *Cd* Caudate nucleus, *Pu* Putamen.

striosome compartment have strong direct projection to the evaluative midbrain dopaminergic cell groups (VTA, SNc), whereas MSNs from matrix project only to non-evaluative BG nuclei (GPe, GPi and SNr)[13,136], and they communicate more within microcompartment borders than across[21,137,138]. Based on these circuit differences and stronger representation of reward-predictive, expected-outcome-related signals in striosomes compared to matrix[139-141] we adopt the microdomains based actor-critic framework[142,143], expected to generalize across the VS, DMS, and DLS macrodomains.

## Matrix

MSN ensemble activity within the matrix compartment is hypothesized to represent the currently engaged behavioral option. Houk, Adams, and Barto proposed that matrix modules function as an actor, which generates signals that command actions or represent plans[142]. In actor-critic architectures, the actor is defined as "a conventional action–selection policy, mapping states to actions in a probabilistic manner"[144]. Similarly, Joel and colleagues suggested that small populations of striatal matrix neurons are capable of eliciting actions[145], based on the actor model proposed by Suri and colleagues[143,146]. Ito and Doya rephrased the proposal that the matrix compartment, particularly within the dorsal striatum, implements the "actor" that learns action selection[100,142,145]. They further suggested that matrix neurons learn "action values" of candidate actions, contributing to model-based action selection and flexible, goal-directed behavior[147-150]. It is important to acknowledge that the literature supporting the specific role of the matrix compartment in representing

engagement with actions are primarily theoretical and computational. To our knowledge, direct experimental evidence specifically isolating the matrix compartment's contribution to action engagement is currently lacking. However, as Crittenden and Graybiel noted, the matrix comprises 80-85% of the dorsal striatum's volume, suggesting that the majority of MSNs reside within this compartment[151]. Therefore, it is highly likely that existing data supporting the role of dorsal striatal MSNs in action selection and execution[34,36-38] predominantly reflect the activity of matrix MSNs. Future research should prioritize directly assessing the specific contributions of matrix and striosome compartments to behavioral engagement and other aspects of goal-directed behavior.

## Striosomes

Striosomes represent evaluative signals about the currently engaged behavioral option. Striosomal neurons fire more to reward-predictive activities, reward-predicting cues and encode more information about expected outcome during learning[139,141]. Further, striosome compartments consistently exhibited stronger representations of reinforcement outcomes than matrix[139], and mediate value based learning[140]. In line with this evidence one version of the actor-critic model posits that striosomes realize the critic role by integrating and providing reward predictive information ("state value"), and its derivatives like RPE, to the actor[142,145,150,152], through a dopaminergic loop (see also later sections). In actor-critic architectures, "The critic corresponds to a conventional state-value function, mapping states to expected cumulative future reward"[144]. Whether striosomes only influence matrix-based engagement with behavioral options, through the midbrain

dopaminergic cell system and learning and relearning, or whether there are other mechanisms, is an open question and requires more research; there is evidence suggesting that MSNs communicate less across striosome-matrix borders then within[21,137,138].

An alternative actor-critic model assigns the critic role to the VS or nucleus accumbens, and the actor role to the DS, rather than to striosomes and matrix, respectively[153–155]. However, several lines of evidence support the striosome-matrix division favored by this review. First, the significantly higher striosome/matrix ratio in the VS compared to the DS[21,156,157], could confound the identification of critic-related activity in the VS and actor-related activity in the DS. Second, a hierarchical engagement framework, as proposed here, benefits from separate critics for goals and strategies. Because even the optimal strategy may not achieve the desired goal on every trial, the predicted value of the strategy will often be lower than the predicted value of the goal. Therefore, independent evaluation of goals (by VS striosomes) and strategies (by DMS striosomes) allows for more precise adaptive learning.

## Different associative mechanisms underlie striatal engagement at different macro-domains

What associative mechanisms can underlie striatal engagement at different macrodomains? Existing data support a framework in which MSN ensemble activity itself can represent the timing of commitment to behavioral options, in a way that multiple populations of MSNs –including cells which change their activity throughout the duration of commitment[129–131] and cells which increase their activity at the onset or offset of it[131,132]. Specific actions are represented continuously in neural activity space by overlapping, rather than discrete, MSN ensembles[33]. Striatal circuit architecture enables the emergence of winner-takes-all (WTA) MSN ensembles representing engaged options; these circuit motifs include fast-spiking interneuron-based[158,159], and MSN lateral inhibition-based suppression of competing MSN ensembles[160,161], as well as iSPN-mediated feedforward suppression of competing BG-output nuclei ensembles[162]. The WTA ensemble activity is driven by convergent synaptic inputs from various cortical and subcortical regions, among which cortical is the largest and thalamus is the second largest source[163,164], highlighting the importance of cortico-striatal and thalamo-striatal inputs in determining moment by moment WTA ensembles[66–72]. Convergent inputs combined with synaptic plasticity make the striatum a powerful substrate for associative learning and switching between engaged options[165]. This learning process establishes a mapping between specific input patterns and behavioral options. Within this framework, we define engagement as the instantiation of a learned association, where a specific pattern of synaptic input drives the postsynaptic activity of a corresponding MSN ensemble. Therefore, 'associative learning' is the process of creating the input-output map, while 'engagement' is the real-time use of that map. With these definitions in mind, we will now examine the associative processes within each striatal macrodomain to elucidate how engagement is realized across the striatum (Fig. 3). The traditional nomenclature, describing associations as stimulus-response (S-R, in DLS), stimulus-outcome (S-O, in VS), or response-outcome (R-O, in DMS), is not consistent from the MSN output point of view for all the macro domains. Especially the R-O association in the DMS, having the outcome as the last element of the association would not explain well the strategy engagement related MSN activity. We suggest that a more biologically grounded approach should emphasize the output of the striatum, the MSN ensemble activity itself. Therefore, we propose a revised nomenclature where the last element in the abbreviation represents the postsynaptic MSN ensemble activity, and the preceding elements represent the presynaptic inputs that drive that activity.

## Ventral striatum: engagement with Goals via S-O and C-O associations

The VS has established roles in representing stimulus-outcome (S-O) associations in both rodents and primates[166,167], in which sensory stimuli serve as predictors of specific outcomes. We propose to augment this framework by incorporating context-outcome (C-O) associations within the

repertoire of VS function. This addition reflects the influence of broader environmental contexts on outcome prediction, enabling a more comprehensive neuronal representation of goal-relevant information within the VS. The activity of VS MSNs represents the outcome (O) – the goal – and this activity is driven by inputs representing either specific stimuli (S) or the broader context (C). For example, VS activity is necessary for initiation of stimulus based engagement with goals in rats[113,114], and 40% of VS neurons discriminate between different expected rewarding liquid outcomes based on visual cues in primates[168]; this discrimination can also be interpreted as a neuronal representation of engagement with specific goals. Multiple further studies corroborate the role of VS MSNs in encoding the properties of desired outcomes[103,115,116,169–172], supporting the idea of S-O and C-O associations. Here, an outcome is defined as a specific external event (change in the environment) contingent on a chosen action —for example, a rewarding object (food pellet, coin), a state change of the environment (gate opening), or sensory feedback (auditory cue); whereas a goal is the active, internal representation of that outcome when it is currently being pursued.

## Dorsomedial striatum: engagement with strategies via O-C-S and O-C-RT associations

The DMS is proposed to support engagement with strategies to achieve selected goals. Its role in adapting strategies is demonstrated by deficits in reversal learning and strategy switching following its inactivation[125]. DMS neurons encode information for maintaining a strategic approach, such as action-values in probabilistic tasks and stimulus-values in conditioning paradigms[6,100,126,127]. We argue this is mediated by outcome-context-stimulus (O-C-S) and outcome-context-response type (O-C-RT) associations, which better capture the information processing required for flexible, rule-based behavior than traditional R-O models. In this revised nomenclature, DMS MSNs represent either the key stimulus (S) in sensory-guided tasks or the response type (RT) in probabilistic tasks, driven by inputs conveying the desired outcome (O) and strategy relevant elements of the current context (C). This is supported by findings that stimulus-responsive MSNs are concentrated in the DMS during visual decision-making (the last element in the O-C-S associations), and that DMS neurons represent advantageous response types during probabilistic tasks (the last element in the O-C-RT associations)[6,100,173–175].

## Dorsolateral striatum: engagement with actions via S-R and A-A associations

Finally, the DLS mediates engagement with actions, controlling their sequence, timing, and duration according to the engaged strategy[95]. DLS neurons systematically encode the identity and temporal structure of action syllables, and represent the selected action itself, independent of its execution speed[33,34].

This function is consistent with stimulus-response (S-R) associations, where MSN ensemble activity represents action engagement driven by cortical and subcortical stimulus (S) related inputs[66–72]. We further propose that the DLS supports action-action (A-A) associations, a mechanism crucial for chaining actions into skilled sequences. In this scheme, DLS MSN activity representing the current action is driven by synaptic inputs conveying information about the preceding action, a mechanism strongly suggested by the sequential nature of action encoding in DLS neurons[34,36,37].

This revised associative framework, incorporating a potentially non-exhaustive expanded set of association types, can facilitate a more comprehensive understanding of how striatal subregions contribute to the selection, execution and engagement of goal-directed behaviors.

## Different PFC regions support the selection and engagement with goals, strategies and actions

The functional specialization and preferential striatal projection topology of different PFC regions raises the question, which prefrontal areas support the selection and engagement with goals, strategies, and actions? At the highest level of the associative cortical hierarchy, the PFC plays a crucial role in representing actions (in the broadest sense, behavioral options), a

fundamental requirement for organizing goal-directed behavior via mechanisms such as short-term memory and anticipatory sets[176,177]. The PFC provides the neural substrate for executive control and working memory[178–180]; furthermore, it generates goals, predicts outcomes based on environmental context[181–183] and enables flexible, goal-directed behavior guided by an internal world model[3,30,31,184–187]. Despite the widespread neuronal representation of task-relevant variables throughout the PFC during goal-directed behavior[188–190], considerable research supports functional specialization within the PFC. This specialization is also evident in the topographic projections from different PFC subdivisions to distinct striatal macrodomains, which suggests a division of labor for selecting goals, strategies, and actions[84] (see also Table-1, and the Box 1; the latter illustrates, through specific behavioral paradigms, how multiple prefrontal regions, operating within serially connected PFC-BG loops, support the hierarchical selection of behavioral options).

The framework presented here aligns with a unified cortico-thalamic system view rather than treating these structures as separate functional entities, based on their extensive bidirectional connectivity[191,192], complementary circuit motifs[191], long-range trans-thalamic cortico-cortical relay function, and broadcasting hub operations[193]. One particularly interesting line of evidence shows that PFC exhibits recurrent excitatory circuit connectivity[194] and high-dimensional, mixed-selective representations of the environment[195,196]. Complementarily, feedforward circuit motifs (without recurrent-excitation) enable the thalamus to extract abstract, low-dimensional representations from these high-dimensional PFC representations[197–201], and broadcast this abstracted information to other cortical areas or back to the originating regions. Further evidence suggests that the thalamus may play an even more prominent role in rule representation[202], based on earlier emergence of abstract rule encoding in thalamus compared to PFC (though it should be noted that only two, not all potential PFC subregions were investigated simultaneously). Further arguments for integrative cortico-thalamic function are findings on communication through coherence[203–205], where the thalamus can play a pivotal role in orchestrating such coherence[206–210]. Therefore, in this review thalamus is viewed as part of an integrated cortico-thalamic system. Thalamic relaying of prefrontal computations to striatum has also been suggested[211]; however, striatum itself also exhibits circuit motifs without recurrent excitation[159,212] and has been suggested to perform dimensionality reduction from cortical representations[145].

## Goal selection
The ventro-medial prefrontal cortex (vmPFC), including the medial OFC, ventral-PrL-, IL-, and frontal pole- cortex (with some interspecies differences) are implicated in goal selection and engagement. For example, Holton and Kolling demonstrated that the vmPFC supports goal commitment in human subjects[213]. The medial OFC plays a critical role in goal selection based on value comparisons, potentially by transforming representations of value into a common currency to facilitate comparisons among diverse options[214–216]. The vmPFC is crucial when affective responses are influenced by conceptual information about specific outcomes[188,217]. This ventromedial region of the PFC works closely together with the VS in the process of goal selection, as evidenced by anatomical connectivity[97,101,102] and can provide outcome-predictive contextual (C) information that drives VS MSN activity in the proposed C-O associations and goal engagement. Furthermore, striosome-biased projections from these vmPFC regions to the VS (further discussed later) are hypothesized to provide evaluative feedback, influencing striatal learning and updating of goal representations. Also based on their functional data Tang and Averbeck proposed that learned value information representing engaged behavioral goals is maintained throughout the ventral cortico-striatal circuit, which includes the OFC and VS[84] (see also Box 1, example C).

## Strategy selection
Strategy selection associated parts of the PFC include dorsal PrL-, lateral/ventral OFC-, and ACC cortices[2,4,218]. Ventral/Lateral OFC is associated with sensory cue based prediction of outcomes and outcome-properties, and in turn in cue-based strategies[181,182,219]; whereas mPFC has been linked more with choosing actions based on outcome history[181,220]. Activity of PrL neurons encodes subsequent trial choice during outcome evaluation periods, and optogenetic inactivation of PrL cortex impairs optimal outcome-probability-based decision-making[221]. mPFC neurons persistently encode probabilistic value based decision variables[222]. Both mPFC and dorso-medial striatal neurons represent advantageous strategy (response-type) in a probabilistic decision-making task, a small proportion of those were putative monosynaptic connected mPFC-DMS neuron pairs[6]. The mPFC is also important in the case of complex rules and strategies which can often be considered as internal model based decision-making[30,223,224]. Lateral/ventral OFC is important in various forms of sensory cue based goal-directed behavior[3,31,218,225]. OFC projections to the DMS are required for appropriate economic decision-making[1] (see also the Box 1 example A and B).

## Action selection
Higher-order motor cortices, including secondary motor cortex and likely the neighboring dorsal part of the ACC (dorsal area 24, ACAd), are primarily associated with selecting and planning upcoming actions[85,226–229]. These regions, in turn, preferentially target the DLS to finalize selection and time the engagement with specific actions. In general, due to methodological constraints, most studies have only recorded and analyzed neuronal responses in one or a few adjacent frontal regions during the same behavior. Only recent improvements in methodology and analysis have begun to enable investigation into differences in information representation across larger sets of regions[230]. This will refine our understanding of inter-regional differences with respect to goal-directed behavior.

## Different PFC inputs drive striatal micro-compartments
What PFC inputs drive striatal micro-compartments? The PFC likely influences striatal engagement through two primary pathways, mirroring the functional distinction between striosomes and matrix. First, striosome-mediated learning allows the PFC to shape striatal associations –routing of specific inputs to specific outputs– over time by providing evaluative feedback (Fig. 4A, B). Second, direct matrix modulation allows the PFC –and other cortical and thalamic inputs– to rapidly switch between engaged behavioral options by altering the pattern of activity in matrix MSNs (Fig. 4C). These two pathways provide complementary mechanisms for both long-term adaptation and short-term flexibility.

## Striosome-mediated learning pathway
Specific PFC regions project preferentially to striosomes within specific striatal macrodomains, providing evaluative signals that drive learning and adaptation. These signals likely represent model-based reward prediction information—that is, predictions about future outcomes based on an internal model of the environment, rather than simply on past rewards[100]. Such a pathway allows the PFC to influence which associations are learned and strengthened within the striatum, shaping long-term behavioral biases.

Observations of striosome-biased connectivity from distinct PFC regions suggest an organization which can support functional specialization. Specifically, a striosome bias from a particular PFC region to a given striatal macrodomain may only be present where the respective PFC region provides critical evaluative signals. Projections from the same PFC region to other striatal areas, where the evaluative role of that PFC region is not critical, may not exhibit striosome bias.

Supporting this concept, Ragsdale and Graybiel found that fronto-striatal projections correspond to a large extent with striatal microdomains[231], more specifically PrL, parts of ACC and posterior OFC send dense innervation to striosomes in the dorsal striatum, with less dense labeling in the surrounding matrix[232,233]. Injections into motor cortices and parts of cingulate cortex showed preferential termination in the matrix[232].

## BOX 1 | Applying the hierarchical engagement framework to specific behavioral paradigms

A) Attentional set-shifting paradigm[325–328] (Rodent example): In this paradigm, the subject must learn rules based on sensory dimensions (e.g., odor or medium) to obtain a single outcome type (food). *Loop 1 (Goal)*: The vmPFC/OFC computes the environmental context and valuation of the potential outcome. VS ensemble activity maintains engagement with the goal and thus motivation for the task. *Loop 2 (Strategy)*: mPFC/OFC predicts/tracks the subject's position in task-state-space based on an internal world model, detects task-state changes and strategy/set-shifts (e.g., "Odor A is no longer rewarded; switch to digging medium A"); and in turn it biases the DMS to engage the currently advantageous strategy (e.g., "Focus on odor"). DMS ensembles represent the engagement with this specific rule or attentional set. *Loop 3 (Action)*: Motor cortices and adjacent A24 compute the current advantageous response based on the engaged strategy and current trial cues (e.g., "Strategy says odor A is correct; odor A is on the left; therefore, go Left"). The DLS maintains engagement with this chosen motor response and its constituent action syllables. (Note: In these cited studies, functional evidence is primarily derived from lesion effects in mPFC/OFC and DMS. The specific contributions of VS and DLS to this paradigm are inferred from the wider literature; see main text)

B) Probabilistic Choice / Bandit Tasks[6,84,222] (Rodent & Primate example): Subjects choose between multiple options (e.g., 2 or 3 ports/targets) where reward probabilities change over time, requiring continuous value updating. *Loop 1 (Goal)*: The vmPFC/OFC computes the expected value of the chosen option. VS ensemble activity encodes and maintains this value signal throughout the trial, representing a temporal commitment (engagement) to the goal that drives continuous motivation. *Loop 2 (Strategy)*: The mPFC tracks the action-outcome history (internal model) to detect changes in reward contingencies; and in turn it biases the DMS to engage the currently advantageous strategy (e.g., "Shift preference to Left"). DMS ensembles encode the specific action-values (e.g., Value of Left vs. Right) required to select the optimal strategy. *Loop 3 (Action)*: Motor cortices and lateral PFC compute the immediate motor kinematics. DLS/DS activity represents engagement with the specific execution of the chosen movement (e.g., reach direction or port choice), emerging later in the trial than VS goal signals. (Note: This entry synthesizes findings across species: Primate studies[84] provide direct evidence for the dissociation of Goal (VS) and Action (DS) engagement, while rodent studies[6,222] elucidate the Strategy/Action-Value updating in the mPFC-DMS loop.)

C) Economic decision-making, goods-based[215,216,329] (Primate example): In this paradigm, in each session there are two predicted appetitive outcome types (fruit juices, offered in different quantities). *Loop 1 (Goal)*: vmPFC/OFC computes environmental context and sensory-cue-based prediction, valuation, and comparison of potential outcomes/goals; VS ensemble activity represents engagement with the chosen goal for each trial (e.g., choose 2 × 65 μL of grape juice over 3 × 65 μL of diluted cranberry juice). *Loop 2 (Stimulus-selection)*: mPFC/OFC maintains the chosen-value signal and outcome identity, and sends this information to bias DMS toward the sensory cue associated with the desired outcome; based on the engaged goal, DMS ensemble activity represents engagement with the sensory cue associated with the desired outcome (e.g., bright green squares for grape juice). *Loop 3 (Action)*: Motor cortices and adjacent A24 compute the current advantageous response based on the engaged sensory cue and current trial spatial configuration (e.g., engaged sensory cue is bright green squares, and bright green squares are on the left side, therefore choose the left-side response), and send this information to bias DLS toward the advantageous response; based on the engaged sensory cue, DLS ensemble activity represents engagement with the current trial's chosen response according to the current trial cues (e.g., choose the left side as the left cue—bright green squares—is the one associated with the desired outcome), and in subsequent hierarchical sub-loops, the moment-by-moment engagement with the current action-syllable within the engaged response. (Functional evidence was obtained from OFC via electrophysiology and optogenetics; but VS, DMS, DLS, mPFC, and motor cortex/A24 were not recorded in the cited studies, so the goal-engagement (Loop 1 striatal component), stimulus-selection (Loop 2), and response-execution (Loop 3) stages of the framework remain to be directly tested in this paradigm, and have separate support from other studies, see main text.)

D) Combined fixed- and free-choice probabilistic decision-making[100] (Rodent example): In the fixed-choice task, specific frequency tones instructed the rats to poke a designated left or right hole for a fixed probability of reward. In contrast, the free-choice task utilized a white noise cue that provided no directional information, requiring the rats to select a side based on reward probabilities that varied across blocks. *Loop 1 (Goal)*: VS ensemble activity showed the strongest activation at task initiation, encoding state value and task type (fixed vs. free-choice context). This represents engagement with the goal of obtaining reward and the general incentive value of the current behavioral context—setting the stage for downstream action selection. *Loop 2 (Strategy)*: DMS ensemble activity most strongly encoded action values during the action selection epoch—the expected value of choosing left versus right given current reward contingencies; DMS action-value encoding was more prominent during free-choice trials (when the animal must compute and compare values) than fixed-choice trials (when the cue directly instructs the response), consistent with DMS biasing selection toward the currently advantageous choice based on the engaged goal. *Loop 3 (Action)*: DLS activity encodes the action command (Left vs. Right identity), peaking just before execution. This signal is prominent even when choices are repeated, consistent with the DLS maintaining the specific motor engagement. (Note: Functional recordings were obtained from all three striatal macrodomains (VS, DMS, and DLS) that provide direct evidence for the hierarchical gradient across striatal stations. PFC regions were not recorded, so the cortical contributions to each loop remain to be directly tested in this task, and have separate support from other studies, see main text.)

In more detail multiple tract-tracing studies demonstrated that cortical regions with striosome-biased striatal connectivity do not project to all parts of the striatum in the same striosome-biased way; rather only to certain areas of the striatum[232,233]. Similarly, Waugh and Blood found that striosome-like voxels with connectivity dominated by one cortical region are organized into distinct zones[234].

Ragsdale and Graybiel observed a similar organizational principle in cats, where corticostriatal projection patterns could be ordered along a single axis according to the dorso-ventral position at which the transition from striosome-biased to striosome-avoidance occurred[235]. This axis included projections from the posterior parietal cortex, dorsomedial PFC, vmPFC, insular cortex, and rostral temporal cortex. In combination, the observed striosome bias in corticostriatal projections can reflect a functional organization, where different cortical regions contribute distinct evaluative information to specific striatal domains.

The medial OFC, IL cortex, and ventral parts of the PrL cortex strongly project to the nucleus accumbens of the VS[97,101,102]. These brain structures have been associated with functions consistent with goal selection, suggesting that striosome bias from these cortical areas may be present primarily in striatal macrodomains involved in goal selection and subsequent strategy selection, such as the VS, but not in the DMS and DLS. Similarly, the dorsal PrL cortex and ventral ACC strongly project to the DMS[102]. These

regions have been associated with functions consistent with strategy selection, suggesting that striosome bias from these cortical areas may be present primarily in striatal areas involved in strategy selection, such as the DMS, but not in the nucleus accumbens or DLS. Finally motor cortices (M1 and M2) and dorsal ACC strongly project to the DLS[98,101,102]. These regions have been associated with functions consistent with action selection[33,34], suggesting that striosome bias from these cortical areas may be present primarily in striatal areas involved in action selection, such as the DLS, but not in the nucleus accumbens or DMS.

### Direct matrix modulation

In addition to the slower, learning-based influence via striosomes, the PFC can also exert direct control over striatal activity through projections to matrix MSNs[14] (Fig. 4C) or connections to striatal interneurons[55]. These projections can convey information about the current state of the internal model, allowing the PFC to rapidly switch between engaged behavioral options[3,30,31,56,57]. This rapid modulation is critical for flexibly adapting behavior based on the current state of an internal model, without waiting for slower, dopamine-dependent plasticity. For instance, projections from motor cortices (M1/M2) to the DLS matrix are well-positioned to directly influence the selection and engagement with actions[85,226–228], while projections from associative PFC regions to the DMS matrix could rapidly update an engaged strategy based on new contextual information[3,30,31]. Importantly, the transition between engaged options does not only rely on PFC as a central controller. Instead, switching is an emergent property of prefronto-BG systems cooperation, where PFC provides internal-world-model state-based predictions as biasing signals about best behavioral options to striatum which executes winner-takes-all selection and engagement of best option[212,236]. When state-transition is detected and best option predictive biasing signals from PFC are changed, the input drive for the active matrix MSN ensemble decreases, destabilizing the current engagement. This allows alternative options, driven by new contextual PFC inputs, to overcome lateral inhibition and emerge as the new dominant option.

The capacity to maintain engagement with a goal or strategy is constrained by the cognitive cost of the underlying internal model. Working memory processes in the PFC provide the top-down 'context' signals required to sustain activity in specific matrix ensembles. Critically, the capacity to generate and maintain these context-dependent biasing signals is limited by working memory resources. When cognitive load is high—such as during dual-task conditions, complex planning, or situations requiring the simultaneous maintenance of multiple task-relevant representations—the fidelity and stability of PFC-to-matrix signals may degrade[237,238]. This degradation reduces the system's ability to sustain goal-directed engagement and can cause the balance of behavioral control to shift toward previously reinforced, habitual response patterns that are less dependent on active PFC maintenance[239,240]. Thus, the direct matrix pathway represents not only a mechanism for flexible, context-sensitive switching, but also a point of vulnerability where resource limitations can compromise goal-directed control in favor of more automatic, stimulus-driven responding.

### Striosomes exert their evaluative teaching function through a mid-brain dopaminergic loop

How do striosomes exert their evaluative teaching function within the striatum? Projections from striosomes are the major source of striatal input to VTA/SNc[136] and provide a dopaminergic mechanism for evaluating ongoing behavior and updating behavioral option selection in the neighboring matrix compartment. This feedback loop enables the striatum to learn and adapt, ultimately refining the selection of appropriate behavioral options in different contexts.

Striosomes exert their evaluative influence primarily through projections to dopaminergic neurons in the VTA and SNc[136]. These dopaminergic neurons, in turn, project back to the striatum, creating a loop that serves as the basis for modulating engaged behavioral options (represented in the

matrix), through changing synaptic weights to learn or relearn (Figs. 2 and 4). Different striatal regions project topographically to distinct areas within the midbrain dopaminergic system, and the dominant input to these dopaminergic neurons dictates their principal return projections to striatum, creating multiple parallel loops that facilitate learning within each striatal macrodomain[76,78,124,241,242]. Alongside these topographically defined "closed" loops, hierarchical striato-nigral projection patterns also allow for a cascading dopaminergic influence from ventral towards dorsolateral striatal regions, enabling cross-domain communication[76,241,243].

Striosome direct pathway neurons send numerous collateral projections to the GP, SNr, and the SNc. A key distinction between striosome and matrix direct pathway neurons is their projection targets; while both project to the SNr, only striosome neurons directly innervate the SNc[13]. This direct innervation of the SNc by striosomes has been further confirmed using Cre-dependent GFP expression in patch-Cre mice (Sepw1-NP67)[14]. Evans and Khaliq observed a high density of striosome-originating synaptic terminals on SNc dendrites that extend into the SNr[48]. These findings suggest that striosomes exert a powerful inhibitory influence on SNc dopamine (DA) neurons, followed by a period of post-inhibition excitation[48]. In vivo optogenetic studies demonstrate the functional consequences of this influence: activating striosomal projections to the SNc suppresses DA release[53,244,245], and modulates habit formation[53] and movement[245]. Importantly, striosomes can also increase DA neuron activity via two reported mechanisms. The first is a polysynaptic pathway via the globus pallidus and lateral habenula[246,247]. The second, originating from striosomal D2R-expressing neurons, involves a single inhibitory relay in the globus pallidus externa[248]. This demonstrates that striosomal direct and indirect pathway neurons can bidirectionally modulate DA release.

This dopaminergic feedback, originating from striosomal input, drives long-term synaptic plasticity at cortico-striatal synapses[52], shaping which inputs ultimately trigger the engagement with specific behavioral options represented in the matrix. Strong, phasic DA release potentiates concurrently active cortico-striatal synapses onto active MSNs (long term potentiation through spike time dependent plasticity), an effect dependent on D1 receptor activation[49–51,249–252].

Although this review emphasizes the contribution of DA to striatal learning, and the role of cortico-striatal and thalamo-striatal inputs in fast switching; the multifaceted nature of dopaminergic signaling likely allows for more functions. Baseline dopamine through D1 receptors increases excitability of dSPNs[253], which can have a permissive effect on selection and engagement of behavioral options[254]. Even though fluctuations in striatal DA, characterized by varying temporal dynamics (including baseline DA levels, phasic, and tonic release), are suggested to be associated with diverse functions, including motor control, action initiation, movement vigor and motivation[255–258]; both phasic and slower ramping dopamine appear to provide reward prediction error signals for optimized adaptive processes[259]. While DA also exerts immediate short lasting effects on cortico-striatal synaptic strength[260] and modulates the timing of MSN excitability[261] thus sharpening the population of active MSNs, potentially fine-tuning striatal engagement; the primary drivers of striatal engagement are the direct cortico-striatal and thalamo-striatal inputs as they define which specific populations of matrix MSNs are being activated. Evidence suggests that DA signal kinetics and synaptic plasticity mechanisms differ between the DMS and DLS[262,263], and between dorsal and ventral striatum[264]. Wave-like DA dynamics could set model expectations and allow early prediction of upcoming behavioral-output adaptations[262], potentially under direct cortical input influence. However, it remains an open question what diverse functions DA signals support in striatum beyond learning[265,266]. In general, within our framework, one would predict a gradient of slower temporal dynamics in ventral versus dorsolateral striatal regions. Such a gradient would align with the different computational demands of each region—slower integration for goal-related updating versus faster processing for action selection—and is consistent with recent findings[264].

## Multiple evaluative brain systems exhibit striosome-biased projections for integrative control over striatal learning and, in turn, engagement

Why do multiple evaluative brain systems exhibit biased projections towards striosomes? The biased projection of multiple evaluative brain systems to striosomes suggests a crucial role for this microdomain in integrating diverse value signals to guide adaptive learning and behavioral updates. This convergence of information allows the striatum to create a comprehensive representation of the overall value associated with currently engaged and potential behavioral options, considering not just immediate reward, but also long-term goals, emotional context, and potential risks. Human MRI connectivity data presents a striosome-biased input, revealing preferential projections from several PFC regions (frontal pole cortex, medial frontal cortex, OFC, ACC), insular cortex, amygdala, and medio-dorsal thalamus[234]. In contrast, the matrix receives a larger proportion of inputs from secondary motor cortex and visual cortex[14], reflecting its role in representing and executing specific actions and strategies. Multiple brain regions involved in value and valence processing send striosome-biased projections to the striatum, as shown by anatomical tracing studies. These include the mPFC[232], OFC[233], amygdala[267], BNST, hypothalamus, and anterior and reticular thalamic nuclei[14]. Notably, several of these subcortical inputs—particularly the amygdala, BNST, and hypothalamus—are critically involved in processing threat, risk, and emotional salience, suggesting that striosomes serve as a convergence point where potential dangers and aversive outcomes are integrated alongside reward-related information. We hypothesize that the convergence of diverse evaluative inputs onto striosomes allows for a weighted integration of these signals. This integrated value signal, relayed via the striato-nigro-striatal pathway, modulates DA release to drive synaptic plasticity in the matrix, thereby updating the associations responsible for engaging with specific goals, strategies, and actions. Further evidence for this integrative evaluative function comes from distinct dopamine kinetics; unlike in the matrix where aversive cues decrease DA concentration, striosomal DA release is elicited by cues predicting both rewards and aversive outcomes[268]. This bidirectional sensitivity to valence suggests that striosomes are specialized for processing a broad spectrum of motivationally relevant information—including potential risks and threats—enabling the system to update behavioral policies based on both appetitive opportunities and aversive contingencies.

Importantly, the balance of influence between these different inputs to striosomes can shift. Under conditions of heightened emotional arousal, perceived threat, or high reward-punishment conflict, the influence of subcortical structures such as the amygdala and hypothalamus on striosomal activity may become more pronounced relative to the PFC[46,269,270]. This shift can bias the integrated evaluative signal toward risk-averse or emotionally-driven behavioral patterns. For instance, when potential negative outcomes are salient, increased amygdala drive to striosomes may promote learning that favors avoidance or cautious strategies, even when PFC-based cost-benefit analyses might favor approach. In pathological conditions characterized by chronic stress, anxiety, or trauma, this imbalance may become entrenched, potentially leading to maladaptive avoidance behaviors or impulsive reactions driven by threat-detection systems rather than deliberative evaluation. More research is required to determine whether integrative or winner-takes-all mechanisms are present at striosomes at different homeostatic and emotional states. The influence of subcortical evaluative inputs on striosomes is expected to decrease along the ventro-medial to dorso-lateral striatal axis, reflecting a presumed stronger role of emotions and innate drives in goal selection compared to selecting action syllables.

## Conclusion and outlook

This conceptual review synthesizes a framework for understanding prefronto-striatal interactions in goal-directed behavior, centered on the concept of hierarchical engagement (Fig. 1). We argue that the striatum functions as a "switchboard", selecting and maintaining engagement with behavioral options (goals, strategies, and actions) under the modulatory influence of the PFC. VS, DMS, and DLS are proposed to mediate engagement with goals, strategies, and actions, respectively. We suggest a revised associative framework (for instance O-C-S and O-C-RT associations in the DMS, Fig. 3), which may more accurately represent the information flow from synaptic inputs to MSN outputs, for specific behavioral engagement types. The PFC influences striatal engagement through two main pathways: striosome-mediated learning, which shapes long-term associations via evaluative feedback, and direct matrix modulation, which enables rapid switching between engaged behavioral options (Figs. 2 and 4). Striosomes, receiving input from multiple evaluative brain systems, integrate diverse valence signals to guide decision-making. This hierarchical engagement framework provides a perspective on prefronto-striatal function and offers new avenues for investigating the neural basis of goal-directed behavior.

While numerous questions remain at the forefront of PFC and basal ganglia research, one critical task is to determine the precise mechanisms by which the PFC modulates striatal activity through both striosome-mediated learning and direct matrix projections. Including, how information about the identity of and transition between different states of the internal world model is transmitted from the PFC to different striatal cell types. This will require further refinement and combination of methods for the simultaneous recording and manipulation of activity in specific PFC and striatal subregions, including striatal microdomains, during well-defined behavioral tasks. Alongside, the accurate location tracking and reporting of functional data will help to better delineate functional gradients or borders across regions and subregions. An interesting question also concerns the functional relevance for cortico-striatal projections of Area 24 as it overlaps with projection patterns of several PFC areas particularly in primates. Advanced computational modeling will be essential for integrating these data and testing hypotheses about the dynamic interactions within prefronto-striatal circuits. How other evaluative regions like the amygdala, habenula, BNST or hypothalamus compete with prefrontal inputs to modulate goal-directed behaviors, remains less understood, and requires substantial new efforts. Finally, the investigation across these levels on how these circuits are disrupted in disorders of goal-directed behavior (e.g., addiction, OCD, schizophrenia, depression) will not only provide critical insights into the pathophysiology of these conditions but also identify potential therapeutic targets. A deeper understanding of prefronto-striatal engagement promises to advance both fundamental neuroscience and the treatment of debilitating neuropsychiatric disorders.

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

## Acknowledgements

For the purpose of Open Access, the authors have applied a CC BY public copyright license to any Author Accepted Manuscript version arising from this submission. This research was funded in whole or in part by the Austrian Science Fund (FWF) DOI: 10.55776/P35747 and 10.55776/COE16. For open access purposes, the author has applied a CC BY public copyright license to any author-accepted manuscript version arising from this submission.

## Author contributions

A.K. have drafted the manuscript and figures, and J.P. revised the manuscript and figures. All authors revised and contributed to the article and approved the final version.

## Competing interests

The authors declare no competing interests.
