## [Transparent Peer Review file · Communications Biology]

Prefrontal modulation of striatal engagement: a hierarchical framework for goal-directed behavior

Corresponding Author: Professor Johannes Passecker

Version 0:

Reviewer comments:

Reviewer #1

(Remarks to the Author)
Review of COMMSBIO-25-8314

The paper nicely integrates several features of prefrontal corticostriatal anatomy, unified as a hierarchical mode of PFC-basal ganglia circuit control of behavior that is capable of learning and behavior selection. The PFC and striatum are not necessarily involved in distinct processes or behaviors, but rather distributed and hierarchically organized PFC-basal ganglia circuits are recruited to select goals, strategies and actions, with the particular domains of PFC-striatal circuits and specific circuit mechanism (striosome vs matrix) engaged depending upon the current context and mode of behavioral engagement. Here, the striatum is akin to a switchboard for controlling behavior, with the circuits able to learn new behaviors in addition to selecting those which are already established. An integrative, hierarchically organized view of the PFC-BG system has been discussed in some recent literature, but the inclusion of striosomes/matrix here is relatively new. The ideas discussed here would benefit from further development/exploration and incorporation of some newer published findings in these circuits. I do have some concerns about the proposed model that should be addressed by the authors before the manuscript would be suitable for publication:

Major:

1. Since it the point of studying cortico-BG-cortical circuits in rodents is to ultimately understand these brain circuits in healthy humans and patient populations, it would benefit the reader if the rodent-specific anatomy is clearly and frequently mapped onto the homologous regions in primates (PFC, striatum) early on and throughout. Some of this occurs a bit late in the paper (i.e., page 6). Clarity is also very important when discussing BG output nuclei; as the GPi of rodents seems to be involved in PFC-BG loops, but these predominantly involve the SNr in primates.
2. On a related note, regarding the concept of macrodomain here – in the NHP, there is a continuous shift of topographically organized PFC representation in the striatum and thalamus. It might be more comprehensive, and easier to map onto humans, if the macrodomains are defined by their PFC inputs rather than anatomical sub-sections of rodent striatum, especially since it seems like the PFC inputs are one of the major ways striatal sub-regions are different, and thus confer function onto striatal subregions.
3. It should be acknowledged that, although it is often convenient to consider behavior in terms of a discretized categories of automatic/habitual vs goal-directed/model-based, much of our behavior is best characterized as e.g., goal-directed habits. This concept is directly invoked by the model, considering that goals and strategies may be maintained without requiring continuous attention (or one might consider the term “automatically”), and are modifiable with changing inputs.
E.g.,
 - Goal-dependent automaticity exists, with representation of goal action links, which can be activated when contextual cues trigger a goal (Verplanken and Aarts, 1999; Aarts and Dijksterhuis, 2000 as cited in Bargh and Barndollar 1996).
 - Automatic goal pursuit is inherently flexible (Bargh and Barndollar, 1996) and is also strategic in nature because implementations are motivated by intention (Wood and Neal, 2006)
 - It is sensitive to outcome devaluation (e.g., classic “habitual” behavior persists even when it is no longer rewarding.) and thus goal-dependent automatic behaviour does not fit the classic definition of a habit (for review, see Yin and Knowlton, 2006; Graybiel, 2008).”
4. Although mentioned in a few places, there is no discussion around the functional significance of the thalamostriatal

projection. Since the thalamus is at the middle of it all, it would strengthen the model if its role is further explored e.g., when would thalamic neurons be expected to engage the striatum vs only engage their cortical targets? What is the proposed impact of changes in BG output nuclei firing on the thalamus, on the state of prefrontal cortical circuit activity, from a mechanistic standpoint?

5. Figure 1 does not include IL or ACC, where do these map onto the proposed model?

6. To expand the discussion regarding what is known about matrix/striosomes a little bit could enhance the paper. This could briefly describe how their inputs and circuitry are different, relative proportions across the striatum zones (i.e., more in nucleus accumbens than the caudolateral striatum) etc?

7. If the PFC either maintains or switches site of matrix involvement, how does it know which are currently relevant, when to switch? It seems that a homunculus is invoked here. Is there a way to avoid a homunculus in this concept?

8. This proposal places heavy emphasis on PFC for representation and selection of abstract goals. However, recent research has shown that the thalamus is instrumental in organizing high dimensional cortical input into goals and setting up their representation in cortex (Phillips et al., 2025 Neuron). Is this model consistent with these observations?

9. Area 24/ACC corticostriatal projections in primates have substantial overlap with all other PFC regions in the primate striatum. This anatomical feature might be relevant to the model proposed here (functional impact or directions for future study).

10. What is the purpose of a PFC hyperdirect pathway in addition to the PFC projections to the striatal matrix proposed to be important for rapid switching of circuits/behaviors?

11. It would be helpful if the authors spend a bit more time explaining/defining the actor-critic architecture for reinforcement learning. A figure (circuit diagram?) might be helpful as well.

12. While models, by definition, involve simplification, to reduce cognitive selection processes to (1) goal engagement, (2) strategy engagement and (2) action engagement seems rather an oversimplification. Perhaps put forward a spectrum from abstract/affective to more concrete/motor outputs? E.g., one might argue that goals tend to be represented by more rodent DMS compared to the VS, while the rodent VS is involved more with incentive salience. Or perhaps this discrepancy arises from a species difference between rodents and primates? Since rodents do not have a granular dlPFC?

Minor:

13. Does “strong projection” mean a dense projection, or that the individual projections themselves are capable of eliciting postsynaptic response?

14. It would be useful if the authors could better define timescales (e.g., slow vs fast) referenced when discussing striosome vs matrix mechanisms, and influence on PFC circuit activities.

15. Emotional context and potential risks are only mentioned somewhat abruptly at the end of the paper.

16. Please define goal and outcome as their difference feels somewhat blurred when discussing e.g., S-O and C-O associations etc.

17. Although it is acknowledged that direct/indirect pathways have already been thoroughly described elsewhere, it would be interesting if the authors could comment regarding potential specialization of function for the direct/indirect pathways of the matrix vs striosomes.

18. Is it known that the sites featuring high convergence are also sites of striosome? Are the sites of high convergence functionally relevant outside the context of learning?

19. Clarity may be improved if a different term (“behavioral-outputs”) is used throughout, since behavior is often defined as an observable output. Its use to define goal and strategy selection could be unintuitive to a reader.

Reviewer #2

(Remarks to the Author)

Title: Prefrontal modulation of striatal engagement: a hierarchical framework for goal-directed behavior

Author(s): Koszeghy A & Passecker J

Summary: The authors present a novel perspective that attempts to unify disparate subfields within PFC-striatal neuroscience. The authors should be commended for taking on this challenge, and presenting a well-thought out and directly testable account. While I am generally positive about this work, I think its motivation and framing could use refinement. Specifically, I feel that critical concepts in motor control, cognitive effort, and working memory which likely act on their proposed model are not adequately addressed, or at the very least, framed in a way that their exclusion is understandable. Additionally, the authors sample from both rodent and primate studies. While this is commendable, the review, as written lacks a robust synthesis that highlights the similarities and differences that cross species work in this space actively debate. I think it would be beneficial if the authors attempt to reconcile and weigh in on these issues as it speaks to the overall generalizability of their model. That said, I am generally positive about the work and believe that with revision, and the input of the other reviewers, may be suitable for publication.

Main Concerns:

1. One overarching question I have is what problem this perspective is addressing. I think the article makes some compelling points and attempts to link many disparate ideas, but in the process, it also creates new concepts and variants widely acknowledged concepts like S-R associations. While I think it's admirable that the authors are challenging and refining dogmatic concepts, it also raises questions about why this perspective is better. The Introduction launches into a very direct explanation of what we will read about, but it doesn't offer the context or perspective for the reader, or at least for me, to see why this perspective is critical.
2. Anatomical specificity – at the end of page 2, starting with line 51, you define several regions of the rodent PFC in terms of standard stereotaxic atlas convention, but Fig. 1 and subsequent discussion on page 4, starting with line 123, switches to vmPFC and dmPFC. While it's clear from using Fig 1 what brain areas you're referring to, this sudden departure without an attempt to align these different naming conventions is confusing. Further, given ongoing debate over the exact function/delineation of rodent PFC I think it's imperative that a standard and well-defined be used. Preferably, some attempt to align this nomenclature with NHP and human homologs would be useful. Mark Laubach's group and several others discuss this point and I think it would be useful for the discussion presented here (<https://pmc.ncbi.nlm.nih.gov/articles/PMC6220587/>). While I recognize this is done to some extent in the section entitled "Different PFC regions support the selection and engagement with goals, strategies and actions", but I think it would be beneficial to have an earlier section walking the reader through the comparative neuroanatomy and more clearly defining your view of them rather than introducing it in the middle of the text.
3. Related to Concern 1, I think this theoretical account would be greatly strengthened by a more robust and explicit synthesis of rodent and primate work. While these PFC and striatal structures seem to share many similarities across species, this is an active debate, and ideally this theory should readily apply to rodent, as well as primate, behavior. For example, the case study literature presented in the ventral striatum section is interesting, but it would be nice to see evidence supporting this in other model systems as well. Also, in the Goal Selection section on Page 13, there seems to be human and NHP examples, but how this relates to rodent data is unclear. I think making the logic of your synthesis more explicit is critical both for the reader's understanding of your idea, but also in terms of identifying where overlap in findings across species does and does not exist.
4. On Page 4, lines 115-117 you claim that the underlying specific patterns of cortico-striatal and thalamo-striatal inputs that elicit DLS MSN activity likely generalize across all striatal macrodomains. Is there any evidence that this is the case? While I agree that this a reasonable prediction, I think if this is just a prediction it should be made clear that this is conjecture and that future studies need to confirm this to be the case.
5. On Page 5, lines 143-147, you bring up the impact of cognitive effort, and its potential to overwhelm the system. There are many studies that have looked at the role cognitive load plays in modulating action selection. I'm curious why some of this work isn't discussed in more depth. It feels like discussion of how working memory processes might modulate striatal input is pertinent to the model, and while I can sort of see how it might fit into your model in Fig 1, it's also somewhat unclear to me as well.
6. One thing that I struggle with is understanding how these results might apply to an existing behavioral task. Dr. Matt Roesch's group has done a lot of work with a variant of the stop-signal task that might provide a model to apply this theory to, but there are many other groups as well. Explaining how your account might predict/ interpret behavioral and physiological findings within a well-defined/ commonly known task might help to strengthen your overall point that this model helps unify many disparate findings in the PFC-Striatal neuroscience. While I realize that one perfect example might not exist, I think some attempt at presenting this model by using some common or relatable task space is critical for helping the reader realize the utility of this perspective.

Minor Concerns:

1. It would be good to check that abbreviations are defined before being used, and the consistency of their use throughout the manuscript should be double-checked. For example, PFC isn't defined, before it is abbreviated in Line 8 of the Abstract.
2. Page 2, Ln 27: Please write out BG before abbreviating in "PFC-BG loops"
3. Page 5, Ln 132: "Murakami et al., 2017" citation is not formatted liked the other numbered citations

4. Page 6, Ln 157: Remove “her”

5. Page 13, Ln 380: “Bari et al., 2019” citation is not formatted liked the other numbered citations

Version 1:

Reviewer comments:

Reviewer #1

(Remarks to the Author)

The authors have made substantial updates to the manuscript to address the reviewer comments. I have no further issues and am happy with the current state of the paper.

Reviewer #2

(Remarks to the Author)

The author's addressed my main concerns and have updated the text appropriately. The additional figures, and increased inconsistency make their perspective understandable across species. Overall, I have no further suggestions.

We thank the Reviewers for their constructive and thoughtful feedback. We have integrated these suggestions into the revised version of the manuscript. Below, we provide a point-by-point response, with our answers marked in blue.

We have submitted two versions of the manuscript: a clean version and one with all changes highlighted in yellow. One of the key revisions, addressing several reviewer comments, is the inclusion of a summary table that provides an overview of cross-species homology based on function and anatomy.

(Please note: The line numbers in our responses refer to the version with changes in yellow, as the clean version may have been updated to follow the journal's guidelines and are beyond our control)

Reviewer #1 :

The paper nicely integrates several features of prefrontal corticostriatal anatomy, unified as a hierarchical mode of PFC-basal ganglia circuit control of behavior that is capable of learning and behavior selection. The PFC and striatum are not necessarily involved in distinct processes or behaviors, but rather distributed and hierarchically organized PFC-basal ganglia circuits are recruited to select goals, strategies and actions, with the particular domains of PFC-striatal circuits and specific circuit mechanism (striosome vs matrix) engaged depending upon the current context and mode of behavioral engagement. Here, the striatum is akin to a switchboard for controlling behavior, with the circuits able to learn new behaviors in addition to selecting those which are already established. An integrative, hierarchically organized view of the PFC-BG system has been discussed in some recent literature, but the inclusion of striosomes/matrix here is relatively new. The ideas discussed here would benefit from further development/exploration and incorporation of some newer published findings in these circuits. I do have some concerns about the proposed model that should be addressed by the authors before the manuscript would be suitable for publication:

Major:

1. Since it the point of studying cortico-BG-cortical circuits in rodents is to ultimately understand these brain circuits in healthy humans and patient populations, it would benefit the reader if the rodent-specific anatomy is clearly and frequently mapped onto the homologous regions in primates (PFC, striatum) early on and throughout. Some of this occurs a bit late in the paper (i.e., page 6). Clarity is also very important when discussing BG output

nuclei; as the GPi of rodents seems to be involved in PFC-BG loops, but these predominantly involve the SNr in primates.

We thank the reviewer for this helpful suggestion, which aligns with comments raised by Reviewer 2 (C2 and C3). We have addressed these points through several key revisions:

- We introduced Table 1 early in the manuscript (lines 34). This table explicitly describes and compares analogous areas in the rodent and primate PFC, detailing their respective connectivity and functions.*
- We added a dedicated paragraph clarifying cross-species differences regarding the BG output nuclei (lines 44-53), specifically addressing the GPi/SNr distinction mentioned by the reviewer.*

2. On a related note, regarding the concept of macrodomain here – in the NHP, there is a continuous shift of topographically organized PFC representation in the striatum and thalamus. It might be more comprehensive, and easier to map onto humans, if the macrodomains are defined by their PFC inputs rather than anatomical sub-sections of rodent striatum, especially since it seems like the PFC inputs are one of the major ways striatal sub-regions are different, and thus confer function onto striatal subregions.

We thank the reviewer for this insightful suggestion. We agree with the concept of a continuous shift in topographically organized representations and have revised the manuscript to emphasize this gradient (e.g., lines 191-200), we also expanded our citations (Pennartz et al., Hintiryan et al., Gao et al.) to support the notion of finer striatal gradients. However, as the vast majority of existing functional literature uses the triple macro-domain nomenclature (VS, DMS, DLS), we have retained this terminology to ensure clarity and ease of reference for the reader. We have, however, added a statement encouraging future research to adopt more detailed mapping approaches, leveraging the possibilities offered by modern digital atlases.

3. It should be acknowledged that, although it is often convenient to consider behavior in terms of a discretized categories of automatic/habitual vs goal-directed/model-based, much of our behavior is best characterized as e.g., goal-directed habits. This concept is directly invoked by the model, considering that goals and strategies may be maintained without requiring continuous attention (or one might consider the term “automatically”), and are modifiable with changing inputs.

E.g.,

- Goal-dependent automaticity exists, with representation of goal action links, which can be activated when contextual cues trigger a goal (Verplanken and Aarts, 1999; Aarts and Dijksterhuis, 2000 as cited in Bargh and Barndollar 1996).
- Automatic goal pursuit is inherently flexible (Bargh and Barndollar, 1996) and is also strategic in nature because implementations are motivated by intention (Wood and Neal, 2006)
- It is sensitive to outcome devaluation (e.g., classic “habitual” behavior persists even when it is no longer rewarding.) and thus goal-dependent automatic behaviour does not fit the classic definition of a habit (for review, see Yin and Knowlton, 2006; Graybiel, 2008).”

We thank the reviewer for this insightful comment. We fully agree that the binary distinction between 'habitual' and 'goal-directed' control is an oversimplification, and that much of behavior is best characterized as goal-dependent automaticity.

In the revised manuscript, we have:

- *incorporated the suggested references (Verplanken & Aarts, 1999; Aarts & Dijksterhuis, 2000; Bargh & Barndollar, 1996) to substantiate the existence of automaticity at the level of goal representations (lines 170-178).*
- *Also we high-light the difference between “engagement” in the proposed frame work and rigid S-R habits. We emphasize that while engagement allows for automatic execution, it differs from classic DLS habits because it remains sensitive to outcome devaluation via the goal selection loop. As long as the goal remains valuable (and the environment stable), the 'switchboard' can maintain the goal and strategy engagement automatically; however, if the outcome is devalued, the goal selection loop disengages the behavior, demonstrating the flexibility inherent in goal-dependent automaticity (lines: 133-140).*

4. Although mentioned in a few places, there is no discussion around the functional significance of the thalamo-striatal projection. Since the thalamus is at the middle of it all, it would strengthen the model if its role is further explored e.g., when would thalamic neurons be expected to engage the striatum vs only engage their cortical targets? What is the proposed impact of changes in BG output nuclei firing on the thalamus, on the state of prefrontal cortical circuit activity, from a mechanistic standpoint?

We thank the reviewer for highlighting the central role of the thalamus (this also relates to Reviewer 1 Comment 8, so we address these in a unified way). We agree that the thalamus is not merely a relay but an essential active element in cortico-cortical communication, provide sensory information to striatum, and also part of the cortico-BG loops mostly on the returning branch (but we also include a reference for cortico-thalamo-striatal direction). We implemented revisions at three levels:

1) Thalamo-striatal inputs are considered as an important component in determining MSN activity, contributing to selecting active ensembles, and through them engaged behavioral options. We have added more supporting references (lines 136 , 337-341, 395-396) and expanded the text (lines 337-341).

2) We added a new section (lines 430-445) detailing the integral role of the thalamus in cortical function via cortico-thalamo-cortical loops, independent of the basal ganglia.

3) We clarified (line 54) that the thalamus completes the cortico-BG-thalamo-cortical loops. Mechanistically, the BG output nuclei disinhibit the thalamus, allowing it to broadcast the engaged behavioral option back to the cortex. This feedback informs the originating and other downstream cortical areas about the engaged behavioral option.

5. Figure 1 does not include IL or ACC, where do these map onto the proposed model?

Indeed, we the aim of Figure 1 is intended as a simplified schematic to provide a high-level overview of the three loop engagement framework. Consequently, we prioritized conceptual clarity over exhaustive anatomical detail in this illustration. However, to ensure completeness, and address the specific point, the newly added Table 1 lists all specific subregions (including the IL and ACC) along with their cross-species comparisons. For consistency we also refer to the larger anatomical unit v/IOFC on the left panel of Fig1.

6. To expand the discussion regarding what is known about matrix/striosomes a little bit could enhance the paper. This could briefly describe how their inputs and circuitry are different, relative proportions across the striatum zones (i.e., more in nucleus accumbens than the caudolateral striatum) etc?

Thank you for this suggestion, we have incorporated at the beginning of the “striosome-matrix function section” (lines 261-267) the necessary information about the main circuit, input and functional differences as the basis of the functional differences.

This additional information is in addition with the previous segments on this topic regarding circuit differences at lines 590-601, input differences at lines 506-548, and relative proportions at lines 304-305.

7. If the PFC either maintains or switches site of matrix involvement, how does it know which are currently relevant, when to switch? It seems that a homunculus is invoked here. Is there a way to avoid a homunculus in this concept?

We thank the reviewer for raising this point regarding the homunculus in executive control, we address this now in the discussion section (lines 549-556). We agree that attributing the 'decision to switch' solely to the PFC would be imprecise, rather: switching is an emergent property of prefronto-BG systems cooperation, where PFC provides internal-world-model state based predictions as biasing signals about best behavioral options to striatum which executes winner-takes-all selection and engagement of best option (Frank & Badre, 2012; Morita et al., 2016).

8. This proposal places heavy emphasis on PFC for representation and selection of abstract goals. However, recent research has shown that the thalamus is instrumental in organizing high dimensional cortical input into goals and setting up their representation in cortex (Phillips et al., 2025 Neuron). Is this model consistent with these observations?

We appreciate the reviewer highlighting this significant and active area of research. Although a full review of this topic is beyond our scope, we have incorporated a relevant and substantial discussion in the revised manuscript (see lines 431-445). We note that our framework is consistent with Phillips et al. (2025), as we conceptualize goal representation as an emergent property of corticothalamic loops. This perspective integrates the active computational role of the thalamus in selecting abstract task-relevant information. Please see also at the answer for Reviewer1-Comment4 that thalamus is important in our model in multiple points.

9. Area 24/ACC corticostriatal projections in primates have substantial overlap with all other PFC regions in the primate striatum. This anatomical feature might be relevant to the model proposed here (functional impact or directions for future study).

Indeed, in Table 1 we now bring the attention of the readers to this point, and highlight the most relevant references for Area 24 (24a and 24b) projections in rodents and primates. We also include a statement in the direction for future study (lines 733-735).

10. What is the purpose of a PFC hyperdirect pathway in addition to the PFC projections to the striatal matrix proposed to be important for rapid switching of circuits/behaviors?

We thank the reviewer for raising this important point. We agree that the PFC hyperdirect pathway likely plays a critical complementary role in rapid switching (e.g., likely by providing a global 'brake' or 'clear' signal). However, similar to our approach regarding the direct/indirect pathway distinction (see Reviewer 1, Point 17), a detailed integration of subthalamic nucleus dynamics falls outside the primary scope of this framework, which focuses on striatum and its internal domains (macro domains: VS, DMS, DLS, and micro domains: Striosomes and Matrix). To adhere to word count limits and ensure the reader remains focused on the core aspects of the 3 loops / 2 stage model, we have kept this discussion brief. We have, however, added a concise acknowledgement of the hyperdirect pathway's role in the revised manuscript (Lines 45-48 & lines 127-130).

11. It would be helpful if the authors spend a bit more time explaining/defining the actor-critic architecture for reinforcement learning. A figure (circuit diagram?) might be helpful as well.

We appreciate the reviewer's suggestion to clarify this concept. We have revised the relevant section (Lines 270-273 & 296-298) to provide a more precise definition of the Actor-Critic architecture in the context of the BG.

At lines 261-311 of the manuscript there is further elaboration about why the Striosomes as Critics, Matrix as Actors model; Fujiyama et al., 2015; Graybiel & Matsushima, 2023 offers an advantage for hierarchical reinforcement learning; as striosomes are distributed throughout the entire striatum (including VS, DMS, and DLS) and thus each hierarchical level (goals, strategies, actions) can possess its own local critic-actor pair. This architecture allows level-specific credit assignment, a computational arrangement not possible with the single-critic model.

At lines 575-603 of the manuscript there is further elaboration on why Striosomes are uniquely positioned to serve as critics given their direct projections to dopamine neurons in the substantia nigra, contributing to the computation of the RPE teaching signals that drive matrix plasticity.

12. While models, by definition, involve simplification, to reduce cognitive selection processes to (1) goal engagement, (2) strategy engagement and (2) action engagement seems rather an oversimplification. Perhaps put forward a spectrum from abstract/affective to more

concrete/motor outputs? E.g., one might argue that goals tend to be represented by more rodent DMS compared to the VS, while the rodent VS is involved more with incentive salience. Or perhaps this discrepancy arises from a species difference between rodents and primates? Since rodents do not have a granular dlPFC?

We thank the reviewer for these insightful suggestions and agree that the tripartite model is a simplification. To address this, we have made the following revisions.

- We added a section (lines 193-201) discussing the continuous functional gradient in the striatum, moving beyond the rigid three-part division, and cited relevant literature on input-based gradients.*
- Consistent with the reviewer's comment on the rodent DMS, we now clarify that the rodent ventromedial portion of the DMS likely contributes to the goal-selection system. This is supported by strong projections from the mOFC and ventral prelimbic cortices to this region (see also the Table 1, and lines 200-201).*
- Regarding the VS, we propose that the concept of 'goal engagement' provides a more unifying framework that encompasses 'incentive salience.' As detailed in the MS (lines 200-230), we argue that the motivational 'wanting' (incentive salience) attributed to the VS is the behavioral manifestation of the striatum maintaining engagement with a selected goal.*

Minor:

13. Does “strong projection” mean a dense projection, or that the individual projections themselves are capable of eliciting postsynaptic response?

We have adapted the wording, to be more precise (line 201-203). Projection strength is defined in these anterograde tracing studies based on qualitative metrics of presynaptic fiber and terminal density (Maily, Haber et al, 2013; Heilbronner, Haber et al 2016).

14. It would be useful if the authors could better define timescales (e.g., slow vs fast) referenced when discussing striosome vs matrix mechanisms, and influence on PFC circuit activities.

We appreciate the reviewer's request to clarify the definitions of 'fast' and 'slow' timescales within our model. In the revised manuscript, we have explicitly defined these timescales based on their underlying biological mechanisms (Lines 119-121 and Lines 125-128):

We define 'fast' modulation as occurring on the timescale of synaptic transmission and population dynamics. This allows the PFC to rapidly select or inhibit specific striatal matrix

ensembles in real-time (e.g., trial-by-trial switching) without requiring synaptic weight changes. (Milliseconds to Seconds):

We define 'slow' adaptation as occurring on the timescale of synaptic plasticity (e.g., LTP/LTD, which can include protein trafficking and synthesis). This reflects the time required for striosome-mediated dopamine signals to long-term reshape the cortico-striatal weights, updating striatal associations for future automatization. (Minutes to Hours)

15. Emotional context and potential risks are only mentioned somewhat abruptly at the end of the paper.

We thank the reviewer for this observation. We agree that emotional context is a critical factor in adaptive decision-making. However, a comprehensive analysis of emotional processing mechanisms and its interplay falls outside the primary scope of this manuscript, which focuses on a 3-loop 2-layer hierarchical model for goal directed behavior.

Nevertheless, we have expanded on the intersection of emotion driven and goal directed behavior in the revised manuscript; we rephrased sections for increased clarity (lines 676-680; 687-690; 692-700); discussing how competition between these brain systems and uncertainty in model-based predictions can shift striosomes based learning away from goal-directed toward more emotional influence.

16. Please define goal and outcome as their difference feels somewhat blurred when discussing e.g., S-O and C-O associations etc.

We thank the reviewer for pointing out the need to disambiguate these terms. In our framework, an outcome is defined as a specific external event (change in the environment) contingent on a chosen action—for example, a rewarding object (food pellet), a state change of the environment (gate opening), or sensory feedback (auditory cue); whereas a goal is the active, internal representation of that outcome when it is currently being pursued. We have added this distinction in the revised manuscript (Lines 371–374) to clarify S-O and C-O associations.

17. Although it is acknowledged that direct/indirect pathways have already been thoroughly described elsewhere, it would be interesting if the authors could comment regarding potential specialization of function for the direct/indirect pathways of the matrix vs striosomes.

Thank you for this suggestion. we added a brief note on the role of direct- indirect-pathway in matrix based behavioral option selection and engagement (Lines 38-44); and in striosome based dopamine signal control (lines: 599-604). As with the hyperdirect pathway (see Reviewer 1, Point 10), we have however limited our discussion of direct/indirect pathway specialization to given word count constraints and maintain the manuscript's theoretical focus. We believe that an elaborate consideration of these specific projection pathways would divert attention from the more central striosome-matrix distinction for learning and engagement.

18. Is it known that the sites featuring high convergence are also sites of striosome? Are the sites of high convergence functionally relevant outside the context of learning?

Thank you for highlighting that the high convergence of cortico-striatal and thalamo-striatal inputs are not only important for learning associations, but in turn also for engaging and switching between those learnt associations, we have adopted the phrasing accordingly at Lines 340-342.

In general, cortico-striatal axonal arborisation patterns seem to be different between striosome and matrix compartments which may point towards a convergence-difference as well (<https://doi.org/10.1523/jneurosci.18-12-04722.1998>).

19. Clarity may be improved if a different term (“behavioral-outputs”) is used throughout, since behavior is often defined as an observable output. Its use to define goal and strategy selection could be unintuitive to a reader.

We thank the reviewer for the suggestion to improve the accessibility of the manuscript by adopting a more inclusive and less ambiguous term. In response, we have replaced 'behavioral-outputs' with 'behavioral option' throughout the text. This terminology is well-established in Hierarchical Reinforcement Learning, where it describes a closed-loop control policy that extends over time (Sutton et al., 1999). In this context, selecting an 'Option' implies engaging with it and maintaining its execution until a specific termination condition is met. This definition aligns perfectly with our hierarchical framework for actions, strategies, and goals, and is also consistent with previous work by e.g. Botvinick, Niv, and Barto (2009). We have explicitly defined this term early in the manuscript (Line 101-111).

Reviewer #2 :

Main Concerns:

1. One overarching question I have is what problem this perspective is addressing. I think the article makes some compelling points and attempts to link many disparate ideas, but in the process, it also creates new concepts and variants of widely acknowledged concepts like S-R associations. While I think it's admirable that the authors are challenging and refining dogmatic concepts, it also raises questions about why this perspective is better. The Introduction launches into a very direct explanation of what we will read about, but it doesn't offer the context or perspective for the reader, or at least for me, to see why this perspective is critical.

We thank the reviewer for this suggestion to enhance the manuscript's accessibility and impact. Accordingly, we have restructured/reworked the Introduction. We now begin by defining the problem statement and identifying the specific knowledge gaps this review aims to address, before subsequently introducing the proposed framework.

2. Anatomical specificity – at the end of page 2, starting with line 51, you define several regions of the rodent PFC in terms of standard stereotaxic atlas convention, but Fig. 1 and subsequent discussion on page 4, starting with line 123, switches to vmPFC and dmPFC. While it's clear from using Fig 1 what brain areas you're referring to, this sudden departure without an attempt to align these different naming conventions is confusing. Further, given ongoing debate over the exact function/ delineation of rodent PFC I think it's imperative that a standard and well-defined be used. Preferably, some attempt to align this nomenclature with NHP and human homologs would be useful. Mark Laubach's group and several others discuss this point and I think it would be useful for the discussion presented here (<https://pmc.ncbi.nlm.nih.gov/articles/PMC6220587/>). While I recognize this is done to some extent in the section entitled "Different PFC regions support the selection and engagement with goals, strategies and actions", but I think it would be beneficial to have an earlier section walking the reader through the comparative neuroanatomy and more clearly defining your view of them rather than introducing it in the middle of the text.

We thank the reviewer for this valuable feedback regarding anatomical specificity and cross-species alignment. As this point coincides with C3 (and Reviewer 1, C2), we have addressed it through the following revisions:

- *We introduced Table 1 (referenced early at lines 33-34) to explicitly map rodent PFC regions to their primate homology, detailing their connectivity and function. This table aligns the function and also the larger anatomical units (vmPFC, dmPFC) to smaller areas of both rodents and primates; as Figure 1. Emphasizes the 3 loops framework for the hierarchical selection of goals, strategies and actions, and the functional findings in the literature often don't allow for higher resolution than the larger anatomical-units, we stay with labels for those on the figure (accordingly Fig1 left panel label is also changed for consistency).*
- *We incorporated the suggested reference, Laubach et al. (2018), along with other literature on prefrontal and striatal homology (line 36, and Table 1).*
- *Regarding the main text, we largely retained the functional macro-domain nomenclature. We chose this approach because the spatial resolution of many referenced studies does not allow for greater anatomical precision. Furthermore, this terminology emphasizes functional continuity (see lines 426; 191-200; and Table-legend line746), as cellular functional gradients likely cross the sharp borders delineated in atlases. We have, added a statement encouraging future research to adopt more detailed mapping approaches, leveraging the possibilities offered by modern digital atlases.*

3. Related to Concern 2?, I think this theoretical account would be greatly strengthened by a more robust and explicit synthesis of rodent and primate work. While these PFC and striatal structures seem to share many similarities across species, this is an active debate, and ideally this theory should readily apply to rodent, as well as primate, behavior. For example, the case study literature presented in the ventral striatum section is interesting, but it would be nice to see evidence supporting this in other model systems as well. Also, in the Goal Selection section on Page 13, there seems to be human and NHP examples, but how this relates to rodent data is unclear. I think making the logic of your synthesis more explicit is critical both for the reader's understanding of your idea, but also in terms of identifying where overlap in findings across species does and does not exist.

We thank the reviewer for these suggestions. We have addressed the need for better species-specific integration through three key revisions on this point.

- *Ventral Striatum Section: We have incorporated specific references for both rodent (lines 217-220) and primate (lines 217-220) studies.*
- *Goal Selection Section: We have similarly expanded this section to include distinct references for rodents (lines 360; 366; 370) and primates (lines 360; 368; 370). We also corrected an error in the subheading in the original version (Line 358)*

- *Cross-Species Mapping: We added Table 1 to explicitly strengthen the integration and comparison of primate and rodent neuroanatomy including functional evidence.*

4. On Page 4, lines 115-117 you claim that the underlying specific patterns of cortico-striatal and thalamo-striatal inputs that elicit DLS MSN activity likely generalize across all striatal macrodomains. Is there any evidence that this is the case? While I agree that this a reasonable prediction, I think if this is just a prediction it should be made clear that this is conjecture and that future studies need to confirm this to be the case.

We thank the reviewer for identifying the imprecision in our wording. Indeed, it is a prediction and this has now been corrected. The summed weight of cortical synaptic inputs to striatum is ranked number one in all striatal macro domains where as the weight of thalamo-striatal projections is ranked number two in all 3 macro-domains (VS, DMS, DLS), this supports the importance of thalamic inputs in all macro domains, with differences in which thalamic nuclei are the major source (Doig, Bolam, et al 2010; Hunnicutt, et al, 2016), we have added these new references at line 338-339.

5. On Page 5, lines 143-147, you bring up the impact of cognitive effort, and its potential to overwhelm the system. There are many studies that have looked at the role cognitive load plays in modulating action selection. I'm curious why some of this work isn't discussed in more depth. It feels like discussion of how working memory processes might modulate striatal input is pertinent to the model, and while I can sort of see how it might fit into your model in Fig 1, it's also somewhat unclear to me as well.

We appreciate the reviewer's insight regarding the impact of cognitive load and working memory on striatal function. We agree that while we briefly mentioned cognitive effort as a constraint, we did not mechanistically explain how WM limits modulate the engagement framework.

To address this, we have expanded the Discussion (Lines 557-568) to incorporate the Evaluative cost of cognitive load. We now explicitly link high working memory load and uncertainty to disengagement of currently active behavioral option in Matrix due to less clear PFC originating biasing signal towards the specific engaged option, which can allow for engagement with other goals, or habitual behavior to take over. In other words WM maintenance provides the stable context signal required for matrix engagement; when load exceeds capacity, this signal degrades, causing the switchboard to fail or revert to habitual (DLS) control.

6. One thing that I struggle with is understanding how these results might apply to an existing behavioral task. Dr. Matt Roesch's group has done a lot of work with a variant of the stop-signal task that might provide a model to apply this theory to, but there are many other groups as well. Explaining how your account might predict/ interpret behavioral and physiological findings within a well-defined/ commonly known task might help to strengthen your overall point that this model helps unify many disparate findings in the PFC-Striatal neuroscience. While I realize that one perfect example might not exist, I think some attempt at presenting this model by using some common or relatable task space is critical for helping the reader realize the utility of this perspective.

We agree with the reviewer that illustrating the framework through the example of specific behavioral paradigms is useful. Unfortunately there is indeed no perfect example in the literature as of now. However we now include in the revised version of the manuscript a TEXT-BOX (see end of Manuscript) in which we walk through the 3 hierarchical loops framework with some well known behavioral paradigms.

We also added a reference about how the stop-signal-task studies from the Roesch Group support the role of ACC in biasing striatum for action selection (line 480 in the manuscript).

Minor Concerns:

1. It would be good to check that abbreviations are defined before being used, and the consistency of their use throughout the manuscript should be double-checked. For example, PFC isn't defined, before it is abbreviated in Line 8 of the Abstract.

Thank you for these observations, we have now double-checked the usage and referencing of abbreviations throughout the manuscript to be complete and consistent.

2. Page 2, Ln 27: Please write out BG before abbreviating in "PFC-BG loops"

Thank you for these observation, this abbreviation is now introduced before usage.

3. Page 5, Ln 132: "Murakami et al., 2017" citation is not formatted liked the other numbered citations

We thank the reviewer for identifying these inconsistencies, it is now corrected.

4. Page 13, Ln 380: "Bari et al., 2019" citation is not formatted liked the other numbered citations

We thank the reviewer for identifying these inconsistencies, it is now corrected.

5. Page 6, Ln 157: Remove “her”

We thank the reviewer for identifying this error, it is now corrected.